# Chilling accumulation in fruit trees in Spain under climate change

Alfredo Rodríguez[1,2], David Pérez-López[1], Enrique Sánchez[3], Ana Centeno[1], Iñigo Gómara[1], Alessandro Dosio[4], Margarita Ruiz-Ramos[1]

[1]CEIGRAM, Universidad Politécnica de Madrid, 28040, Madrid, Spain.
[2]Universidad de Castilla-La Mancha, Department of Economic Analysis and Finances, 45071, Toledo, Spain
[3]Universidad de Castilla-La Mancha, Faculty of Environmental Sciences and Biochemistry, 45071, Toledo, Spain
[4]European Commission, Joint Research Centre (JRC), Ispra, Italy

*Correspondence to*: Alfredo Rodríguez (alfre2ky@gmail.com, alfredo.rodriguez@uclm.es)

**Abstract.** Growing trees are quite vulnerable to cold temperatures. To minimise the effect of these cold temperatures, they stop their growth over the coldest months of the year, a state called dormancy. In particular, endodormancy requires accumulating chilling temperatures to finish this sort of dormancy. The accumulation of cool temperatures according to specific rules is called chilling accumulation, and each tree species and variety has specific chilling requirements for correct plant development. Under global warming, it is expected that the fulfilment of the chilling requirements to break dormancy in fruit trees could be compromised. In this study, the impact of climate change on the chilling accumulation over Peninsular Spain and the Balearic Islands was assessed. For this purpose, bias-adjusted results of 10 Regional Climate Models (RCMs) under Representative Concentration Pathways (RCPs) RCP4.5 and RCP8.5 were used as inputs of four different models for calculating chilling accumulation, and the results for each model were individually compared for the 2021–2050 and 2071-2100 future periods under both RCPs. These results project a generalised reduction in chilling accumulation regardless of the RCP, future period or chilling calculation model used, with higher reductions for the 2071-2100 period and the RCP8.5 scenario. The projected winter chill decrease may threaten the viability of some tree crops and varieties in some areas where the crop is currently grown, but also shows scope for varieties with lower chilling requirements. The results are relevant for planning future tree plantations under climate change, supporting adaptation of spatial distribution of tree crops and varieties in Spain.

## 1 Introduction

Growing fruit trees is an important source of income for farmers. Spain is one of the largest producers of fruits and vegetables in Europe, with 7437 million euros from 7.4 million t of exported fruits in 2017 (FEPEX, 2018) from a total fruit production of 23.17 million t (MAPA, 2018). With a broad range of climates, Spain produces temperate fruits, subtropical (mainly citrus but also other crops) and even some tropical fruits. In absolute terms, among fruit crops olive trees occupy the largest land area (2.52 million ha), followed by vineyard (0.94 million ha), almond (0.58 million ha), citrus (0.26 million ha) and peach trees (0.09 million ha). According to its agricultural production, Spain ranks first in the world for production of

olives, fourth for peaches and fifth for grapes and pears. In terms of production, one of the most important groups of fruit trees are the temperate trees, accounting for approximately 48% of the total world fruit production according to FAOSTAT (2018). In Spain, fruit trees are concentrated mainly on the east coast, along the river valleys of the coast, especially in the Ebro and Jucar valleys; specifically, apples are found mainly in the North-West and North-East of Spain and peaches are found mainly in North-East and South-East of Spain. Olive trees are concentrated in the south of Spain, especially in the Guadalquivir River valley. Vineyards have a more diffused distribution but are abundant in central Spain (Fig. 1).

Growing trees are quite vulnerable to frost. To minimise the effect of these cold temperatures, they change to a hardy state during the coldest months of the year, stopping their growth and modifying their cells. This state is called dormancy and was defined by Lang et al. (1985) as "any temporary suspension of growth of any structure containing a meristem". These authors also defined different dormancy types depending on the factors that regulate them. In this sense ecodormancy is related to environmental factors, paradormancy to physiological factors outside the affected structure (e.g. apical dominance) and endodormancy is linked to physiological factors inside the affected structure. Since the early 19[th] century, it has been known that endodormancy requires accumulating cool temperatures to be broken (Knight, 1801). Endodormancy is the way fruit trees endure the lowest temperatures of the year and synchronise with environmental factors (i.e. seasonal temperature pattern).

Therefore, the accumulation of time exposed to cold temperatures as experienced by the plant is relevant to estimating the dormancy break date. For this purpose, several models have been proposed to calculate winter chill. The Chilling Hours model is the oldest and the simplest one, quantifying winter chill as the number of hours during the winter season, when temperatures are between 0 and 7.2°C (Bennett, 1949; Weinberger, 1950). The Utah Model (Richardson et al., 1974) uses chilling units and considers that temperatures have a different response depending on the temperature range they belong to, with temperatures above the threshold having a negative effect on chilling accumulation. Chilling portions are the units of the Dynamic model (Fishman et al., 1987a; Fishman et al., 1987b), which accounts for the temporal sequence of cool and warm temperature periods observed in chilling accumulation.

Each tree species and variety has specific chilling requirements for correct plant development, usually related to the environmental conditions where it evolved or was bred. The fulfilment of these requirements can be estimated using different models, such as those mentioned above. As a result, for a given species a range of estimates of chill accumulation encompassing all varieties has to be considered. For instance, for the apricot varieties considered in Campoy et al. (2012), the estimated accumulated chilling varies between 413 ('Palsteyn' variety) and 1172 ('Orange red' variety) chill hours (chilling hours model). This range is 613-777 when chilling units by Utah model are computed, and 37-64 chill portions when Dynamic model is applied.

Given that the driving variable of dormancy start and break is temperature, global warming has to be taken into account in any assessment of future fulfilment of tree chilling requirements. In fact, in absence of significant mitigation measures, global warming is likely to reach 1.5ºC between 2030 and 2052 compared to pre-industrial annual mean global temperature levels (IPCC, 2018). In this respect, several researchers (Campoy et al., 2011; Luedeling et al., 2009a; Luedeling et al., 2011; Luedeling, 2012; Gabaldón-Leal et al., 2017) have pointed out that under this warming scenario, the fulfilment of chilling requirements for some crops and varieties is likely to be compromised. To develop suitable adaptation strategies for both the short and long term, reliable projections of chilling units under different emission scenarios or Representative Concentration Pathways (RCPs, van Vuuren et al., 2011) are needed.

Coupled Ocean-Atmosphere General Circulation Models (GCMs) are a useful tool to provide data for climate change impact models, as demonstrated by recent projects such as the Coupled Model Inter-comparison Project phase 5 (CMIP5; Taylor et al., 2012) or in studies such as Luedeling et al. (2011) in which GCMs were used for analysing the climate change impact on chilling accumulation. However, the coarse horizontal resolution of GCMs (typically 100–200 km) is a significant limiting factor. Due to the increasing demand of policymakers and end users for regionalised projections, the Coordinated Regional Climate Downscaling Experiment (CORDEX) initiative has recently been created (Giorgi and Gutowski, 2015). CORDEX EUR-11 is based on Regional Climate Model (RCM) outputs and provides regionalised projections of key atmospheric variables at 0.11° resolution (~12 km) over Europe among other regions.

Even if RCMs have been proved to be a useful tool to describe climatic features in the tropics (Nikulin et al., 2012; Gómara et al., 2018) and extratropics (Jacob et al., 2014; Casanueva et al., 2015), they still present biases in temperature and precipitation over Europe (Casanueva et al., 2016; Dosio, 2016), overestimating future temperature projections for instance (Boberg and Christensen, 2012). Several techniques have been developed so far to minimise and handle model biases, as future projections of threshold-based indices may not be reliable when models' outputs are used without prior bias adjustment. The bias adjustment technique used here is based on a transfer function such that the marginal cumulative distribution function of the adjusted variable matches that of the observations. A complete discussion of the technique, including validation and effect on climate indices can be found in Piani et al. (2010a), Piani et al. (2010b), Dosio and Paruolo (2011), Dosio et al. (2012), Ruiz-Ramos et al. (2016) and Dosio and Fischer (2018). Through the use of transfer functions, temperature biases from RCMs are often adjusted, showing good performance not only for the central tendency measurements, but also for probabilistic distribution properties over time (e.g. Dosio et al., 2012; Dosio, 2016; Ruiz-Ramos et al., 2016). Dosio (2016) showed that bias-adjustment largely improves the value of present and future threshold-based indices (e.g., the number of frost days): these indices are generally poorly simulated over the present climate, such that the projected climate change may not be reliable. Although it is known that bias-adjustment can affect climate change signal at some extent (see e.g. Casanueva et al., 2018 for the quantile mapping method), these techniques are considered a valid

alternative to apply on climate model outputs to crop models, especially suitable for handling regions of complex orography (Maraun and Widmann, 2018), as it is the case of Spain.

The impact of climate change in global chilling accumulation by using GCMs climate projections was analysed in Luedeling et al. (2011), although the coarse resolution does not allow extracting practical recommendations for Spanish farmers. With higher resolution, Luedeling et al. (2009a), performed an analysis focused on the California region, and there are some other studies in other regions of the world (see Table 3 in Luedeling, 2012), but their results are still difficult to apply for helping in decision making in Spain. In addition, recent studies working with multi-crop model ensembles suggest that ensemble results tend to improve as the number of ensemble members increases. For instance, in Martre et al. (2015) the errors decreased when the ensemble members increased, with little decrease beyond 10 members. This debate was analysed from the statistical point of view in Wallach et al. (2018). More common are the multi-climate model ensembles, as the one used by Gabaldón-Leal et al. (2017), who worked with an ensemble of climate models of 12 bias-adjusted members, and applied the de Melo-Abreu model to analyse impact of climate change for olive trees in southern Spain.

The usefulness of the studies that combine chilling models and climate projections to quantify the future impact relies upon the availability of chill assessments where the chilling requirements of different crops and varieties have been analysed. There are a number of such assessments in a wide range of locations: for example, for estimating the chilling requirements in Murcia, Spain, using the Utah and Dynamic models, for apricots (Campoy et al., 2012; Ruiz et al., 2007; Viti et al., 2010) and for sweet cherry cultivars (Alburquerque et al., 2008), or in Gerona, Spain, using the Dynamic model for apple trees (Funes et al., 2016); and in other places than Spain, several studies have been conducted using the mentioned chilling models (see e.g. Aybar et al., 2015; Benmoussa et al., 2017a; Benmoussa et al., 2017b; Marra et al., 2017; Miranda et al., 2013; Razavi et al., 2011; Sawamura et al., 2017). Even the Chilling Hours Model, which is the oldest method to estimate winter chill accumulation and considers all hours with temperatures ranging from 0 to 7.2 ºC equally effective, is still widely used (see e.g. Londo and Johnson, 2014 and Houston et al., 2018 for grapevine or AEMET, 2018 for analysing the risk of frost in Spain).

The objective of this paper is to assess the impact of climate change on fruit tree chilling accumulation in peninsular Spain and the Balearic Islands, which in turn will strongly affect the viability of different crops/varieties in the 2021–2050 and 2071–2100 future periods. For that purpose, a suite of four chilling accumulation models (each one individually considered and studied) were used by applying the last generation of high-resolution, bias-adjusted climate projections to represent the response of the main tree crops in Spain. To our knowledge, no other previous study provides high-resolution projections of chilling accumulation for the whole peninsular Spain and the Balearic Island by using four bias adjusted climate ensembles (one per chilling model) of 10 members (10 RCMs).

## 2 Material and methods

### 2.1 Observed and simulated climate data sets

The climate variable required by the chilling models used in this study is hourly temperature, which can be derived, when no other information is available, from minimum (Tmin) and maximum (Tmax) daily temperatures. To this aim, available daily observations of Tmax and Tmin for the 1976–2005 period were selected from the Spanish Meteorological Agency (AEMET) weather station records. Missing records up to 10 days were allowed and linearly interpolated to fill the gaps.

Additionally, daily Tmax and Tmin for the same period were taken from the high-resolution observational gridded data set Spain02 (v5, Herrera et al., 2012; Herrera et al., 2016) with horizontal spatial resolution of 0.1° (*ca.* 10 km). This gridded data set is available for downloading (http://www.meteo.unican.es/datasets/spain02) and is freely distributed for research purposes. It  was selected for its high data density and resolution, higher than other observational data sets (e.g. E-OBS, Haylock et al., 2008).

Daily outputs of simulated daily Tmax and Tmin for peninsular Spain and the Balearic Islands were extracted from 10 different CORDEX EUR-11 RCM historical simulations (~12 km horizontal resolution; see Supplementary Table 1S). The 10-model ensemble (hereafter EUR-11) used in this study is based on the availability of model runs (at the chosen resolution) at the time of data processing. This ensemble size is considered to be large enough by the agricultural impact community to retrieve robust results (Martre et al., 2015; Rodríguez et al., 2019). Due to the complex orography of the Iberian Peninsula and its remarkable climatic diversity (Bladé et al., 2010), no additional systematic selection was performed to reduce the number of RCM ensemble members (e.g., Mendlik and Gobiet, 2016). A thorough analysis in this sense would imply decomposing the Iberian Peninsula into several climatic sub-regions (Wilcke and Bärring, 2016) and would derive into a much more complex process to potentially improve already robust results. The outputs of the EUR-11 ensemble for two RCPs were considered: 1) +4.5 W/m$^2$ radiative forcing increase at the end of the 21st century relative to pre-industrial levels (RCP4.5) and 2) the same but for +8.5 W/m$^2$ (RCP8.5).

Subsequently, the Tmax and Tmin data of each member of the EUR-11 ensemble were bias-adjusted, relative to the 1976–2005 Spain02 observation data set, for the historical or baseline period (1976–2005), the  2021–2050  and the  2071-2100 future periods (RCP4.5/RCP8.5) climate conditions (hereafter EUR-11 refers to the temperature bias-adjusted ensemble). A previous bi-linear interpolation was applied to Spain02 0.1° areal-representative grid to match the 0.11° rotated CORDEX grid. The bias-adjustment technique applied has been extensively described and applied in previous studies (Piani et al., 2010a; Piani et al., 2010b; Dosio and Paruolo, 2011; Dosio, 2016; Ruiz-Ramos et al., 2016; Dosio and Fischer, 2018). It consists of a histogram equalisation method that makes use of a two-parameter linear transfer function, which is applied to simulated model outputs. The resulting bias-adjusted data has a cumulative distribution function (CDF) comparable to that of

the observational data set. Detailed information on the technique and scripts used here can be found as supplementary material in Ruiz-Ramos et al. (2016).

The chilling models used in this study require hourly temperature data. The approach initially presented by de Wit et al. (1978) was used to estimate hourly data from daily fields for both observed and simulated data sets. The method estimates the hourly temperature taking into account the sunrise time (previously estimated using the latitude and the day of the year) as well as daily Tmax and Tmin (Reicosky et al., 1989).

## 2.2 Chilling modelling

The EUR-11 climate models' outputs were not directly used by the models but first, the bias adjustment process was performed, and hourly data was prepared from the bias adjusted data. Then four different models were used to estimate chilling over Peninsular Spain and the Balearic Islands: the Utah model (Richardson et al., 1974) originally developed for peach trees, and two of its adaptations (Richardson based models, RbM); the North Carolina model (Shaltout and Unrath, 1983) developed for apple trees, and the model specifically developed by De Melo-Abreu et al. (2004) for olive trees. The Dynamic model (Fishman et al., 1987b), also developed for peach trees, was considered. Models based on the Utah model use chilling units, while the Dynamic model uses chilling portions (in this paper the chilling unit terminology will be utilised in general unless stated otherwise).

The Utah model, which was developed based on Redhaven and Elberta peach varieties grown in Utah, is a mathematical model that calculates the number of chilling units accumulated within several temperature ranges, where optimum efficiency for chilling unit accumulation is within 2.5 and 9.1°C. Temperatures outside that range have lower efficiencies and temperatures above 15.9°C penalise chilling accumulation by subtracting chill units. We used the weights from Richardson et al. (1974), which are the most used, but other versions with modified chilling accumulation values for different ranges exist.

The North Carolina model is an adaptation of the Utah model where temperature ranges have been adjusted for apple trees. Specifically, it was developed for Starkrimson Delicious apple variety under a wide range of temperature and elevations corresponding to five locations in North Carolina (Shaltout and Unrath, 1983). The North of Spain, orographically complex and where most of the national apple production is concentrated, includes regions with similar temperature regimes to which the model was developed (see e.g. Kottek et al., 2006; Peel et al., 2007). An example of a current application of the model can be found in the elaboration of apple frost risk maps (http://www.nrcc.cornell.edu/industry/apple/apple.html) by the Northeast Regional Climate Center from USA administration, implemented by the University of Cornell.

The De Melo-Abreu et al. (2004) model, consisting in a generalisation and simplification of the Utah model applied to olive, showed good performance. Model development was based on 15 olive cultivars grown in four locations (Cordoba and Mas Bové in Spain and Santarém and Elvas in Portugal). It consists on a piecewise function that reaches the maximum chilling unit accumulation when the temperature is optimal. Chilling units linearly decrease as temperature diverges from the optimum, accumulating negative chilling units for high temperatures (penalisation). Recently, this model has been applied in Northwest Argentina (Aybar et al., 2015) with satisfactory results.

The Dynamic model was developed for Israeli weather conditions, and incorporates detailed bud responses to temperature based on experimental data with peach trees (Erez et al., 1990). The model computes the chilling in a two-step scheme. First, cold temperatures promote the formation of a precursor in a reversible process. Second, once the precursor has reached a certain threshold, warmer temperatures promote the irreversible transformation of the precursor into a chilling portion. For the implementation of this model, the equations and standard parameters available in Luedeling and Brown (2011) have been followed (see Code 1S in suppl. mat.).

Results from different chilling models were treated as separated ensembles (when combined with climate projections), and therefore they were not averaged nor directly compared in absolute terms but they were interpreted individually. The chilling period was calculated separately for each chilling model, year, member of the EUR-11 ensemble and grid cell. Fig. 2 describes the process for generating the chilling maps for understanding the whole process followed in each cell, from the self-regulating period calculation, to the aggregation procedure followed for years and climate models. The chilling units' accumulation, built on an hourly basis, was calculated from the moment in autumn at which chilling units started to increase until the moment that it reached its maximum (Fig. 2a). Therefore, the beginning and end of the period vary in each case. This chilling accumulation for the Dynamic model is much easier to calculate because the chilling portions accumulation do not decrease (and for this reason a chilling model different than the Dynamic model was chosen to illustrate the process in Fig. 2). Once the annual chilling sum of a cell was calculated, the 30-year mean value of each member of the EUR-11 ensemble was computed (Fig. 2b). Then the median of the ensemble members was calculated (Fig. 2c). Repeating the process for each cell of the CORDEX grid over peninsular Spain and the Balearic Islands, chilling maps were obtained for each of the chilling models (Fig. 2d).

Chilling model programming, calculations and data processing were done by means of MATLAB software (MATLAB, 2017). Scripts for calculating chilling accumulation with the four models, for the hourly temperature estimation and for the computation of chilling accumulation period are available as supplementary material (Codes 1S to 9S) and can be used under quotation.

**2.3 Data set validation and projection calculation**

To evaluate how the interpolation of daily Tmax and Tmin from Spain02 affects the results, chilling units calculated with 42 AEMET station data and with the closest Spain02 cell, according to the nearest neighbour method, were compared. This evaluation was conducted mainly to check whether the hourly time series derived from the Spain02 data is comparable to the time series of the AEMET stations over coastal and mountainous areas, although the entire grid over Spain was compared. The mean absolute percentage error (MAPE) between AEMET-based and Spain02-based chilling units calculated with the four chilling models was computed for the baseline period.

In the same way, to evaluate the results obtained with the bias-adjusted EUR-11 ensemble for this specific application, the MAPE between Spain02-based chilling units and the median of chilling units from the EUR-11 ensemble was calculated for the baseline period and for the four models.

Then chilling projections were computed with the four models and for the 2021–2050 and 2071-2100 periods for the EUR-11 ensemble. Changes between baseline and future simulated chilling were calculated. Projections were derived from individual RCMs by first averaging each time series (30-year mean of chilling accumulation) and then calculating the ensemble median among the resulting 10 means (one per ensemble member).

Inter-annual variability was measured by the ensemble mean coefficient of variation (CV) of the yearly chilling units of each period (30 years) of Spain02 and the 10 EUR-11 ensemble members. Uncertainty coming from RCMs was measured by ensemble inter-model spread, in turn estimated by the ensemble interquartile range (IQR) of the 10 ensemble members' 30-year means.

**3 Results**

**3.1 Performance under current climate**

Chilling units calculated with Spain02 are in good agreement with those obtained from the corresponding AEMET stations (Fig. 3c, filled dots), with most of the locations with MAPE values lower than 5% for every chilling model. Only a few coastal or mountainous locations presented MAPE values higher than 20%. Therefore, the Spain02 data set was considered acceptable for use as the observational gridded data set to perform the EUR-11 ensemble bias adjustment.

Chilling units calculated with the four chilling models for every CORDEX cell with Spain02 (Fig. 3a) and with the EUR-11 ensemble (Fig. 3b) were in good agreement (Fig. 3c), with MAPE chilling values of EUR-11-based compared to Spain02-based generally lower than 5%. MAPE values were higher than 20% only for some coastal or mountainous regions (those

grid cells are highlighted in diagonal lines in the rest of plots to stress that results should not be considered there). Therefore, the remaining temperature biases after bias adjustment were small enough to enable the bias-adjusted EUR-11 ensemble to adequately reproduce the chilling units' behaviour derived from the observational data set in most locations. The ensemble median for the 1976–2005 period (Fig. 3b) was taken as the chilling accumulation simulated by the EUR-11 ensemble. Inter-annual variability of chilling accumulation was simulated in a similar way when using Spain02 and the EUR-11 ensemble, with small differences for the De Melo-Abreu and Dynamic models in the southern half of Spain (Fig. 1S in suppl. mat.).

As simulated by EUR-11, the four ensembles estimated higher chilling accumulation in the North of Spain, as expected by the cooler conditions in that part of the country. In general, the spatial pattern of chilling accumulation is similar for the different chilling models used (Fig. 3b).

The mean inter-annual variability measured with the CV (Fig. 4a) was, in general, lower than 20% for most of the grid cells whatever chilling model was considered. The Dynamic model presented the lowest results with maximum CV values for some points of the South coast of the Iberian Peninsula. The De Melo-Abreu model showed CV values similar to the Dynamic model with the exception of the mountainous regions where the CV was higher. Finally, the Utah and North Carolina models performed similarly with the CV around 10% in northern Spain and around 20% in southern Spain, with higher values on the South and East coasts.

The uncertainty associated with the EUR-11 ensemble was very low, as the ensemble spread measured by the IQR (Fig. 4b) was lower than 5 chilling portions for the Dynamic model and lower than 100 chilling units for all Utah, North Carolina and de Melo-Abreu chilling models in all the simulated areas except for small mountainous areas.

### 3.1 Chilling projections under climate change scenarios

The median of the four EUR-11 based ensembles show a general decrease of chilling units and portions over all simulated areas for both the 2021–2050 and 2071-2100 periods under both RCPs (Figs. 5b, 5d, 6b and 6d), being more pronounced by the end of the century, as expected. Under RCP4.5, a decrease of up to 30 chill portions in the Dynamic model based ensemble and up to 600 chill units for the rest of the chilling ensembles is projected for the 2021–2050 period (Fig. 5b). A slightly higher but similar decrease is projected under the RCP8.5 scenario in the 2021–2050 period (Fig. 5d). In the 2021–2050 period under both RCP scenarios, and in 2071-2100 period under RCP4.5 scenario, the change was fairly spatially homogenous (Figs. 5b, 5d and 6b). However, in 2071-2100 period under RCP8.5 scenario an agreement among the chilling model ensembles points that the largest chilling accumulation changes are projected for the North and North-West coast and for the South-East coast of Spain, with decreases larger than 60 chilling portions calculated with the Dynamic model and larger than 1200 chilling units calculated with the Utah, North Carolina and de Melo-Abreu models (Fig. 6d).

CV values (Fig. 7) are similar for the 2021–2050 period between RCP4.5 and RCP8.5 and for all chilling model-based ensembles, with the Utah and North Carolina based ensembles presenting higher CV values than the De Melo-Abreu and Dynamic ones. In the 2071-2100 period, both climate model inter-annual variability and uncertainty (CV and IQR, respectively, see Figs. 7 and 8), in general, increase with respect to the NF period for every model ensemble; both are higher in the RCP8.5 scenario. As found in the NF period, the Utah and North Carolina chilling model-based ensembles presented higher CV values than the De Melo-Abreu and Dynamic ones. The IQR obtained is larger for the 2071-2100 period and RCP8.5 scenario, as expected.

To illustrate the consequences of these projections, we can analyse the mean number of compromised seasons in four representative productive locations for some common varieties of apple, olive and peach trees (Table 1). All the crops and varieties used in the example would be severely compromised in the 2071-2100 period and RCP8.5 scenario at the selected locations, so adaptation would be mandatory for that period. However, at Murcia, adaptation would be required from now on. On the other side, the analysis shows how some varieties (peach Sunlite and Flavortop at Buñol, East of Spain) may offer resilience until mid of the century.

## 4 Discussion

The chilling portion results are in agreement with the projections from Luedeling et al. (2011) in the Mediterranean region for different periods, where emission scenarios and global climate models were averaged (see Fig. 6 in Luedeling et al., 2011; in addition information for determining the impact of climate model and emissions scenario was provided in that study). Our work shows the spatial distribution of a generalised decrease in chilling sums projected for the rest of the 21$^{st}$ century. Gabaldón-Leal et al. (2017) used an adapted version of the De Melo-Abreu model to calculate the projected chilling units for olive trees in the Andalusia region, also showing a generalised decrease in chilling accumulation projected for the rest of the century.

Our work has similar findings than other studies performed in other parts of the globe; for example, Darbyshire et al. (2013), who analysed the impact of the future warming on winter chilling in Australia concluding that adaptation will be necessary in many locations, at least at some extent, within the next 50 years. Also, a negative impact of climate change on chilling accumulation was found, as expected, in another region with Mediterranean climate, California, in Luedeling et al. (2009a), showing that adaptation would be difficult for some crops under some scenarios. This would be also the case in Brazil (Wrege et al., 2010). According to these studies, adaptation for some tree crops appears to be much more difficult in other parts of the world than we found in this study for Spain.

The projections of the chilling accumulations provided in this study have a lower uncertainty coming from simulated climate scenarios (as indicated by IQR values) than the common uncertainty levels of impact assessments (e.g. Lorite et al., 2018; Tao et al., 2018). This is probably because these chilling models are based only on temperature, and there is higher agreement in the climate change signal related to mean temperature increases than for other climate variables. When other

climate variables are required for impact assessment, the uncertainty is usually higher (e.g. Olesen et al., 2007). For olive trees, previous studies indicate that the lack of knowledge on crop chilling requirements may introduce much more uncertainty than climate projections (Gabaldón-Leal et al., 2017). In any case, according to the validation process, the high-resolution bias-adjusted CORDEX data provide temperature values with adequate quality for this particular application. However, it is important to stress that in spite of the relatively low IQR values shown here (except for mountainous and

coastal areas), in certain places these temperature values were approximately 50% of the value of the change.

Special attention should be paid to the south-western zones of Andalusia, with a substantial oceanic influence, and coastal locations of the Mediterranean, where the evaluation of the selected data sets did not perform as well as for the rest of the country and where tree crops are significant. In light of the results, our hypothesis is that the meteorological stations in these

areas are poorly represented by the interpolated Spain02 data set. In addition, some authors have noted part of these zones as potential areas of crop extension for climate change adaptation (Gabaldón-Leal et al., 2017 for olive trees). In these areas precisely, inter-annual variability (as indicated by CV values) appears to be quite large, which may pose an additional challenge, especially in the 2071-2100 period and for the warmer scenario. These high CV values, particularly for Utah and North Carolina models, could be related to the low values of chilling accumulation in absolute terms, which might become

these models too sensitive for warm places.

Uncertainty was also higher in some mountainous regions, where chill increases are found for warmer climate projections for both the Utah and de Melo-Abreu model-based ensembles. At first glance contradictory, this is explained by the temperature thresholds used in the models and is in agreement with the results reported by Luedeling et al. (2011) who found that

warming from a cold baseline (with temperatures so low that they do not contribute to the chilling sum) can lead to winter chill increases, while warming from a warmer baseline should lead to chilling decreases. Nonetheless, few tree crops are grown in these areas.

Ideally, a site-specific calibration would be desirable for any simulation exercise. However, when data are not available for

the targeted area, a common practise is to extend models' application to locations where the conditions are similar with those where the model was calibrated. For example the Utah model is applied without site-specific calibration in locations different than Utah (e.g. Alburquerque et al., 2008 for cherries in Spain, Razavi et al., 2011 for peach and apricot in Iran, or Sawamura et al., 2017 for peach in Japan). However, it is important to stress that these models were developed, more than for specific locations, for specific tree species (see materials and methods section). In addition, a current practice is to use the

chilling models against phenological data of a specific species, generally for several varieties, and obviating that the model was fitted for a different crop, assuming that there are not differences among species (e.g. Alburquerque et al., 2008; Benmoussa et al., 2017a; Benmoussa et al., 2017b; Elloumi et al., 2013; Funes et al., 2016; Prudencio et al., 2018). In our work, we have prioritized the adjustment parameters made to the Utah model (Richardson et al., 1974) for different species (apple, which became North Carolina model; and olive, which became de Melo-Abreu model), under the hypothesis that the model would perform better if fitted to the behaviour of each species than if the model was used with the parameters established for peach.

However, uncertainty from the chilling models themselves remains. There are several studies (e.g. Benmoussa et al., 2017a; Luedeling et al., 2009b; Ruiz et al., 2007; Zhang and Taylor, 2011) indicating that the Dynamic Model (DM) exhibits a higher accuracy than the RbM, but at the same time, the reported improvement in those studies is very small (e.g. Ruiz et al., 2007), and they also report varieties and locations where RbM models perform better. Also, some studies claim that there is not a significant difference between models performance; for instance, in Alburquerque et al. (2008) differences between DM and Utah model were not found when estimating chilling requirements for cherry cultivars in Spain, or in Ruiz et al. (2007) chilling requirements of the evaluated apricot cultivars were very homogeneous according both to the Utah and Dynamic models, also in Spain. Both Utah and Dynamic models still have room for improvement in terms of accuracy as found (among other chilling models) in Luedeling (2012). In the same study, the DM was recommended for warm regions. However, DM did not show good performance in the Tunisian warm climate Sfax region estimating chilling requirements for pistachios (Benmoussa et al., 2017b), and despite of providing better chilling estimates than other models for almonds in the same Sfax region, the DM showed some shortcomings indicating that is not well adapted to that climate (Benmoussa et al. (2017a). In view of all these evidences, we cannot conclude that DM is a better model for Spain in general terms and therefore four different chilling models were considered in this study.

To improve the existing models, and while functional understanding of dormancy process progresses (Campoy et al., 2011), more experimental data, whose availability is limited in Spain, should be generated to improve the chilling simulation, not only because of the differences between locations, but mainly due to the huge uncertainty related to the species and variety requirements. Targeted field experiments should be designed for this purpose.

The method used in this study to compute the chilling sum period every year was crucial to increase the quality of our projections, since the expected warmer temperatures for the Iberian Peninsula will definitely affect the onset and duration of such a period. A fix period could become eventually meaningless in Spain in a climate change context, causing inconsistencies when the cold period is clearly shifted at the end of the century. This self-regulating calculation period approach has been considered for chill models because of the lack of reliable physiological markers and the inefficacy of fixed dates to account for the seasonal climate variability (Measham et al., 2017). According to Marra et al. (2017) a self-

regulating algorithm approach to calculate the starting date, similar than the one used here, allowed a significant improvement compared to a fix date. Also, results in Measham et al. (2017) show a larger variability in the chilling portions accumulation using a fix date approach than a self-regulating one, as some chilling portions were excluded due to a late initial date. However, some studies found chilling responsive periods not covering the full winter season (e.g. Guo et al., 2015; Luedeling et al., 2013). Nevertheless, having in mind this possible chilling accumulation overestimation, the use of a self-regulating method makes the chilling projections comparable across periods (i.e. baseline, 2021–2050 and 2071-2100 periods) and RCPs. Thus, the computation period has evolved dynamically over the 21$^{st}$ century for every climate and chilling model, RCP, moment in the century and location considered.

According to our results, it is expected that some areas where temperate trees are currently grown will not be suitable in the 2071-2100 period for some crop varieties depending on how rapidly greenhouse gasses emissions evolve. The main difference between the two studied RCPs, consisting in a larger reduction of chilling accumulation in RCP8.5, is projected to be more accentuated in 2071-2100 period, as expected. According with the chilling requirements and the example on the number of compromised seasons (see Table 1), this severe scenario could cause, for example, that the broadly cultivated Golden Delicious apple variety which requires 1050 chill units (measured with the North Carolina model), had difficulties in fulfilling its chilling requirements under a RCP8.5 scenario in the Ebro valley (see Fig. 2Sa in suppl. mat.), where these apples are currently grown. For olive trees, although MAPE validation values at the southeastern-most part of the Andalusia region presented the highest values, reasonable doubts can be raised on the viability of important olive tree varieties such as Picual, with requirements of 469 chilling units (estimated with the De Melo-Abreu model) in that region in the 2071-2100 period under the RCP8.5 scenario, according to the projected chilling accumulation values (see Fig. 2Sb in suppl. mat.). This result is in agreement with the reduction in the suitable cultivation areas in Andalusia for this variety, as found by Gabaldón-Leal et al. (2017). In the case of the widely grown Redhaven peach variety, with requirements of 813 chilling units (estimated with the Utah model) and 73 chilling portions (estimated with the Dynamic model), in the 2071-2100 period under the RCP8.5 scenario, a lack of chilling requirement accumulation is projected by both Utah and Dynamic model-based ensembles according to the projected chilling accumulation values (see Figs. 2Sc and 2Sd in suppl. mat.), at the South-West and East of Spain, also including large regions of Murcia, Valencia and the Balearic Islands, which are zones with a currently high peach production. The high number of compromised seasons would lead to a change of variety in these areas, which despite of these important impacts, would be enough in most locations to adapt to the projected chilling accumulation under future conditions. In general, for all tree crops considered here, there are some varieties with low chilling requirements with adaptation potential; for example Anna apple variety (218 chilling units measured with the North Carolina model, Hauagge and Cummins, 1991), Aprilglo peach variety (8 chilling portions and 150 chilling units measured with the Dynamic and Utah models respectively, Erez, 2000) or Arbequina olive variety (339 chilling units measured with the de Melo-Abreu model, Table 1).

Further work to advance towards more accurate projections of chilling sums, while new experimental data are generated, would be to analyse the confidence that the chill requirements of most important crops and varieties in Spain are fulfilled, as well as for the variety with the lower chilling requirement within a given species. This would enable us to analyse the chances of local adaptation, given that matching chill sums and varieties must be done at the local scale. The analysis would be a refinement of the current impact assessment, and it should be done not only in terms of mean or median results from the different climate models, but also providing additional measures of robustness, as 1-year events can have long-lasting consequences on tree crops. This becomes a relevant issue when analysing ensemble scenarios with high associated uncertainty (e.g. 2071-2100 period or RCP8.5). It could be possible to use a hypothesis-based index such as the ensemble outcome agreement index (e.g. EOA, Rodríguez et al., 2019) to test the robustness of a hypothesis that imposes a conservative threshold, for example, considering the threshold for a variety to meet the "safe winter chill" requirements at a specific location and time (Luedeling et al., 2009a). By doing this, suitable zones for a given variety could be calculated.

Finally, this study is yet another call for action, to carry out not only adaptation but also mitigation measures, to limit the warming rate within the 1.5°C as claimed by the last IPCC special report (IPCC, 2018). The present results strongly support that local adaptation would be much more feasible for moderate warming scenarios (RCP4.5 and below) than for RCP8.5.

## 5 Conclusions

A generalised reduction of chilling accumulation is projected across peninsular Spain and the Balearic Islands regardless of the climate scenario, future period and chilling calculation model used. The reduction is projected to be higher for the 2071–2100 period and for the RCP8.5 scenario as expected.

A winter chill reduction may threaten the viability of some crop varieties, especially in some areas that already have a low number of chilling units and are cultivated with chilling demanding species, where their reduction may jeopardise the cultivation of some varieties within the 2021–2050 period.

An improvement of chilling projections was accomplished here by combining high-resolution RCM outputs, bias-adjusted against a gridded observational data set and contrasted with station data, and then applying four chilling models using an evolving chilling period onset. At our current knowledge, such an assessment by four independent ensembles has not previously been done. The uncertainty related to these projections coming from climate data is lower than in other impact assessments, while further studies are needed to improve our knowledge on chilling requirements and modelling for a wide range of tree crops and varieties.

Finally, this climate change impact should be considered for future tree crop plantation and choice of variety, and also for designing adaptation strategies; these results enable local adaptation by helping to match chill sums and varieties over the 21st century. Such an adaptation would benefit from mitigation, as adaptation is assumed to be more feasible for moderate warming scenarios.

## 5 Author contribution

The conceptualisation and methodology design were done by MRR and AR. The tree physiological aspects were supervised and written by DPL and AC. ES supervised the methodology and analysis process related to climate data processing. The bias-adjustment technique was developed by AD. IG produced the bias-adjusted data. AR developed and applied all the code scripts used to generate the results, including the figures. AR and MRR prepared the manuscript and all co-authors reviewed 10 it and contributed to the final version.

## Competing interests

The authors declare that they have no conflict of interest.

## Acknowledgements

We acknowledge the World Climate Research Programme's Working Group on Regional Climate, and the Working Group 15 on Coupled Modelling, the former coordinating body of CORDEX and responsible panel for CMIP5. We also thank the climate modelling groups (listed in Table 1S of this paper) for producing and making their model output available. We also acknowledge the Earth System Grid Federation infrastructure, an international effort led by the U.S. Department of Energy's Program for Climate Model Diagnosis and Intercomparison, the European Network for Earth System Modelling and other partners in the Global Organisation for Earth System Science Portals (GO-ESSP). The authors thank AEMET and UC for the 20 data provided for this work (Spain02 v5 data set, available at www.meteo.unican.es/datasets/spain02). Iñigo Gómara was supported by the Spanish Ministry of Economy and Competitiveness (Juan de la Cierva-Formación contract; FJCI-2015-23874) and Universidad Politécnica de Madrid (Programa Propio – Retención de Talento Doctor). Alfredo Rodríguez was supported by Spanish National Institute for Agricultural and Food Research and Technology and Agencia Estatal de Investigación Grant MACSUR02- APCIN2016-0005-00-00 and by the Comunidad de Madrid (Spain) and Structural Funds 25 2014-2020 (ERDF and ESF), project AGRISOST-CM S2018/BAA-4330.

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

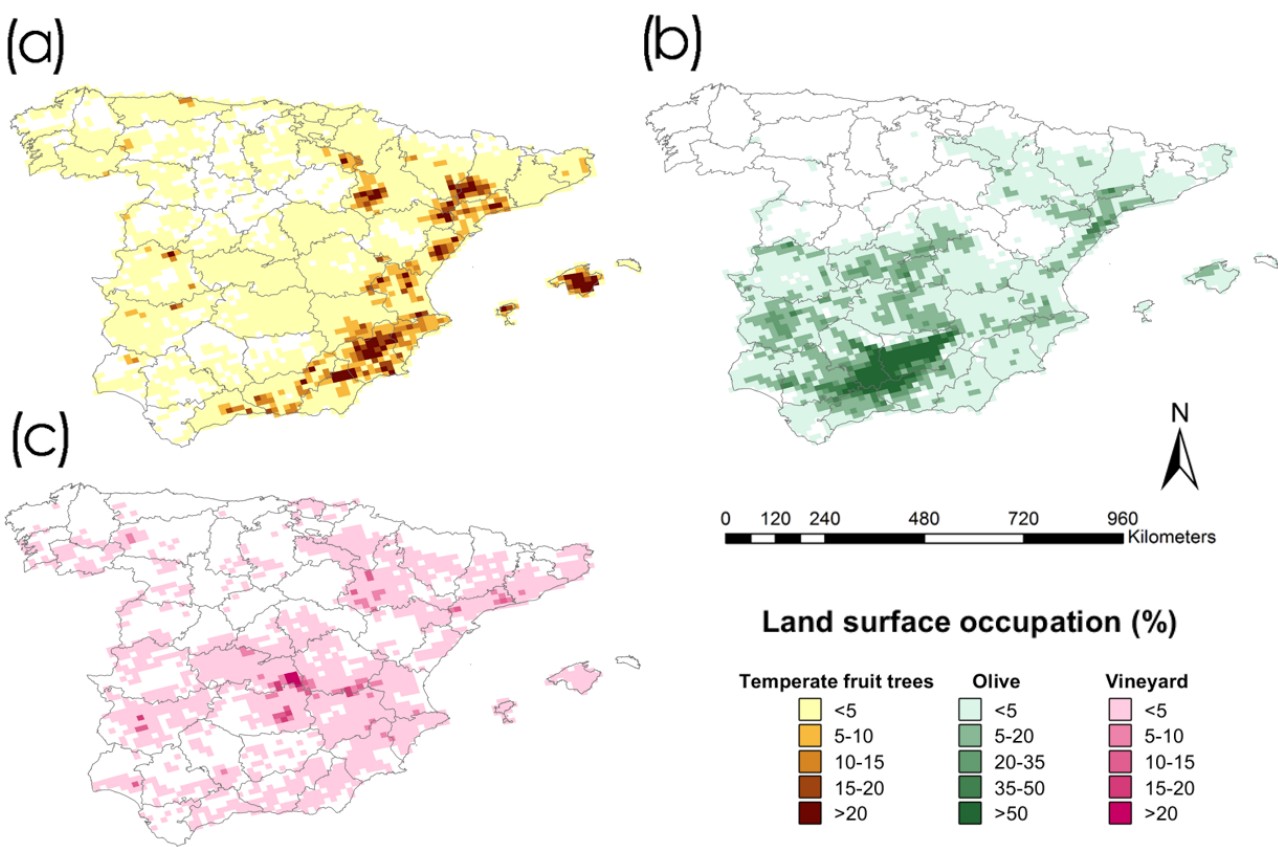

Figure 1. Percentage of land surface occupation (from the total cell area) for temperate fruit trees (orange colour tones, plot a), olive (green colour tones, plot b) and vineyard (pink colour tones, plot c) in 2011. Maps were created with the information available from the Spanish Soil Occupation Information System (SIOSE, 2015)

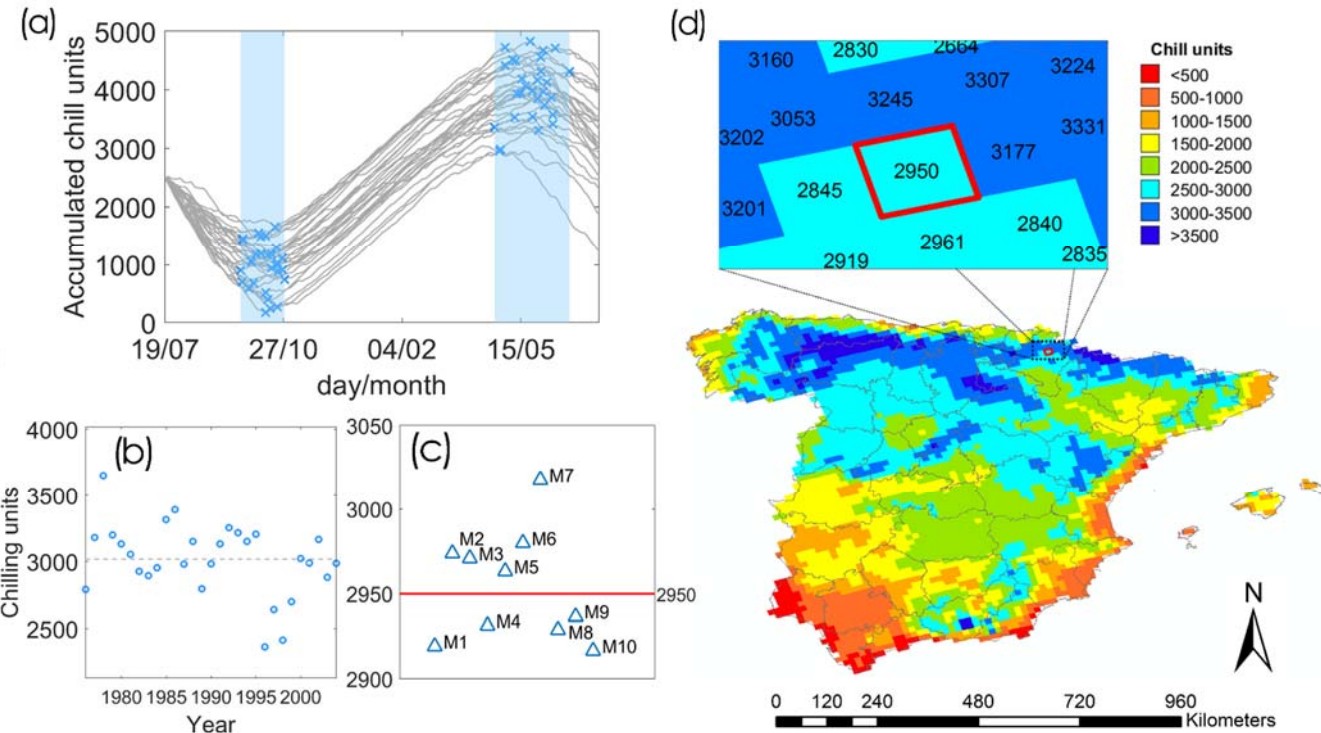

Figure 2. Methodological example: Chilling accumulation calculation process using the North Carolina chilling model in a particular location (42.86ºN, 1.57ºW) for the 1976–2005 period, for a single location and chilling model. Plot a) shows the initial and final chilling accumulation dates for each year (blue crosses) of the EUR-11 ensemble member IPSL-CM5A-MR/RCA4 and the date spread (vertical blue band). Plot b) shows the annual accumulated chilling units (blue dots) and the model mean (dashed horizontal line). Plot c) shows every ensemble member's mean (M, blue triangles) and the ensemble's median (red horizontal line). The results for each grid cell are then used, for each RCM, to create the map shown in plot d).

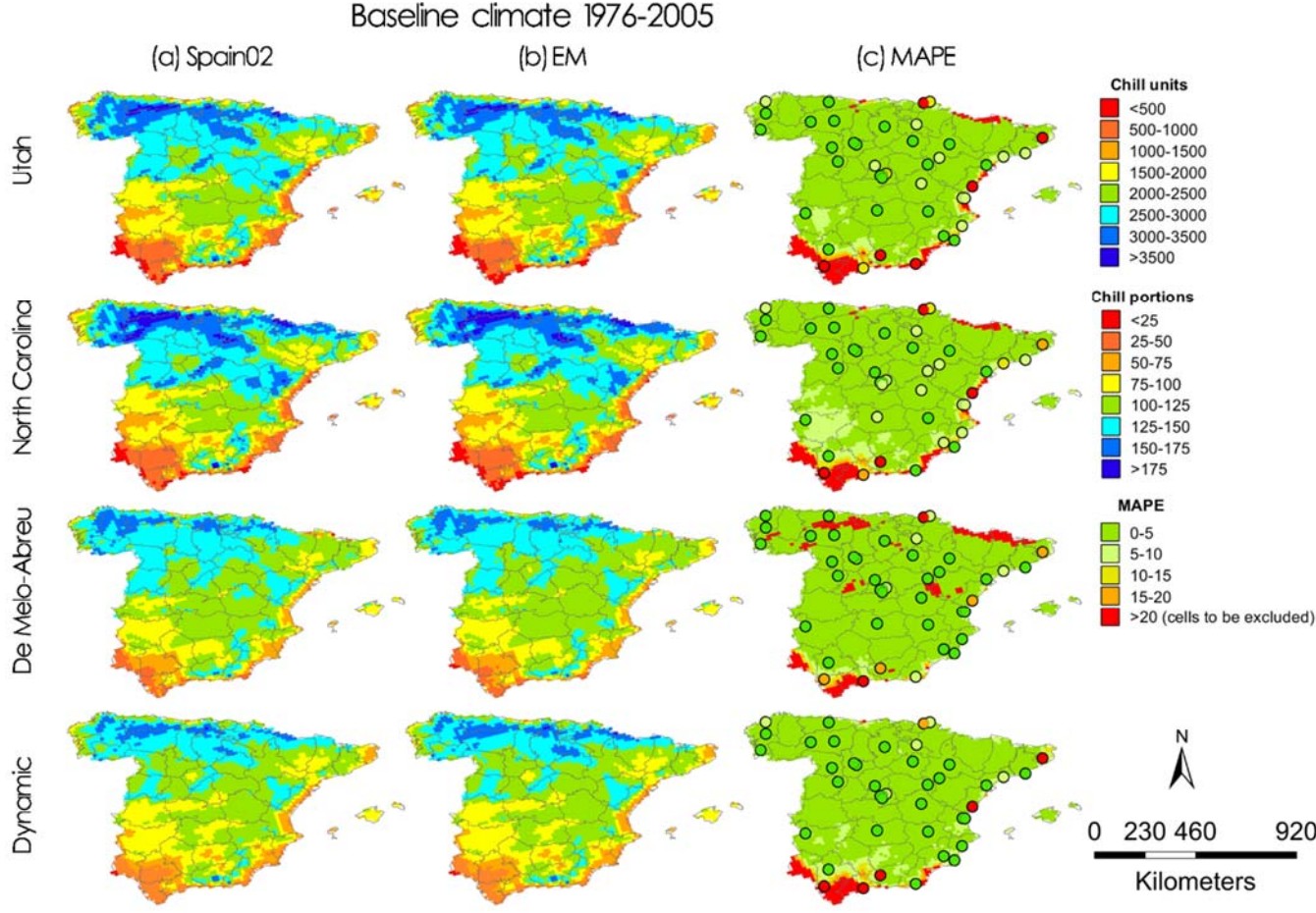

**Figure 3. Evaluation of the chilling accumulation units for the baseline period (1976–2005). For each chilling model (rows), the map of the 30-year mean chilling sums calculated with the observational data set Spain02 (first column) and the map of medians of the 30-year mean chilling sums of the 10 EUR-11 ensemble members (EM, second column). Mean absolute percentage error (MAPE, %) between chilling accumulation calculated with Spain02 and EM data sets (third column, map) and between those calculated with Spain02 and AEMET data sets (third column, dots). Cells with MAPE values greater than 20 (in red) are not considered as reliable results. 95th percentile MAPE values for each model were 25.8 (Utah), 23 (North Carolina), 25.9 (de Melo-Abreu) and 14.2 (Dynamic).**

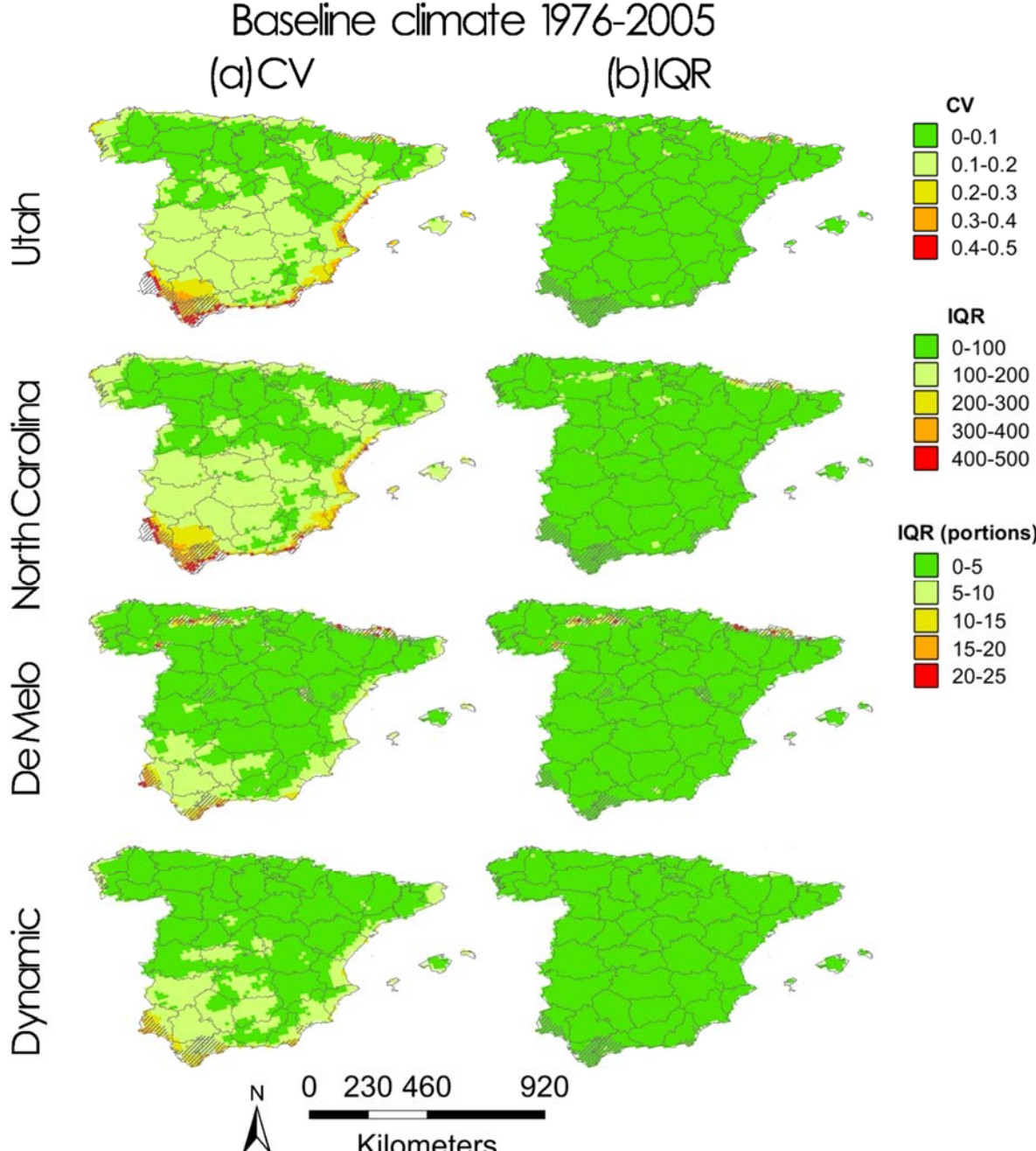

**Figure 4. Maps of inter-annual variability and uncertainty results for the baseline period (1975–2005). For each chilling model (rows), the 10 EUR-11 ensemble members' mean coefficient of variation (CV, expressed per unit) of the 30-year period (first column) and the ensemble's interquartile range (IQR, measured in chill units or chill portions respectively) of the 30-year mean chilling sums of the 10 EUR-11 ensemble members (second column). Grid cells with mean absolute percentage error (MAPE, %) values from the validation phase higher than 20 are not considered as reliable results and are highlighted in diagonal lines.**

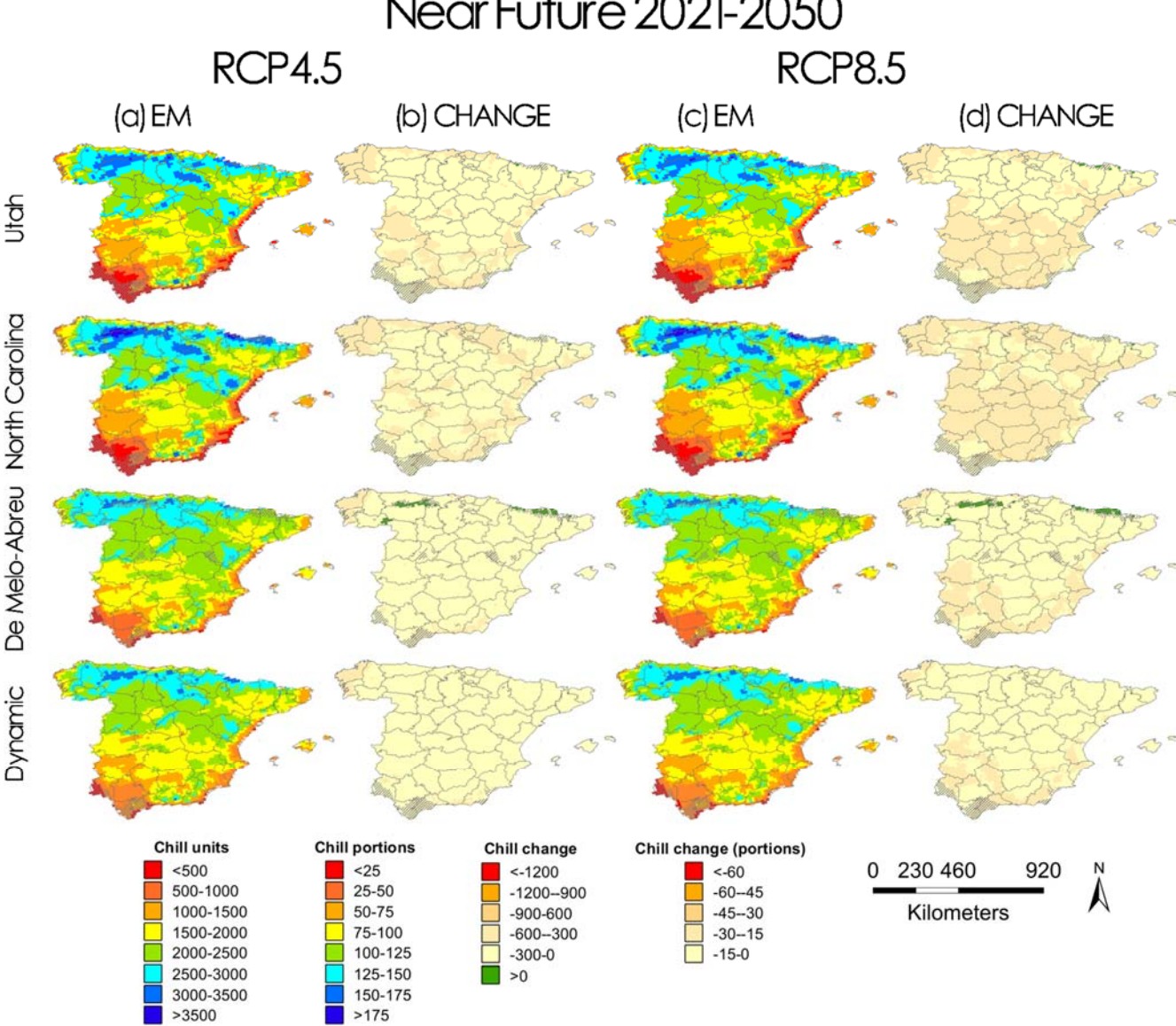

**Figure 5. Maps of chilling accumulation in the 2021-2050 period. For each chilling model (rows), the median of the 30-year mean chilling sums of the 10 EUR-11 ensemble members and change in chilling accumulation with respect to the baseline period, for RCP4.5 scenario (first and second column respectively) and RCP8.5 scenario (third and fourth column respectively). Grid cells with mean absolute percentage error (MAPE, %) values from the validation phase higher than 20 are not considered as reliable results and are highlighted in diagonal lines.**

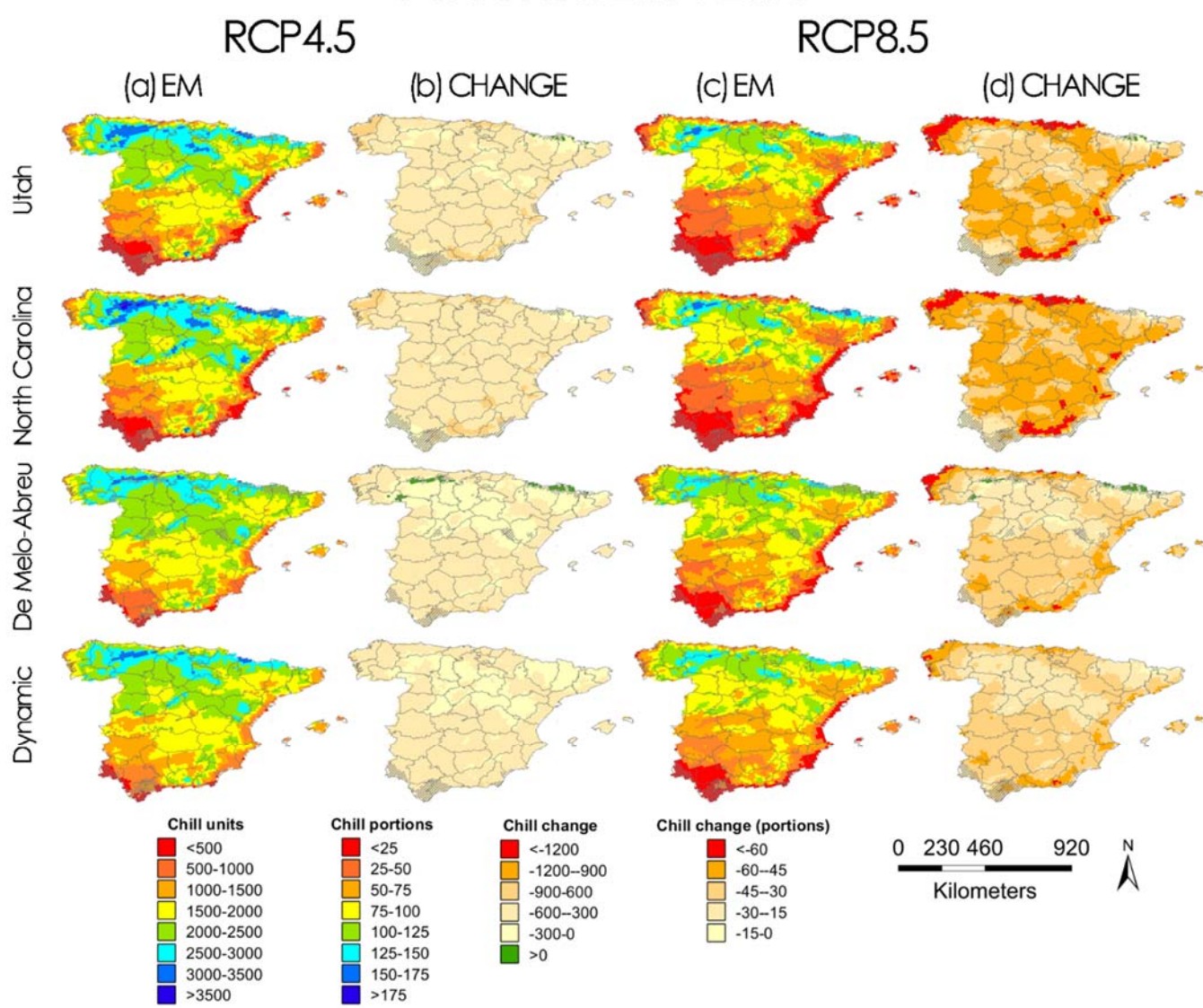

**Figure 6. Same as Fig. 5 but for the 2071-2100 period.**

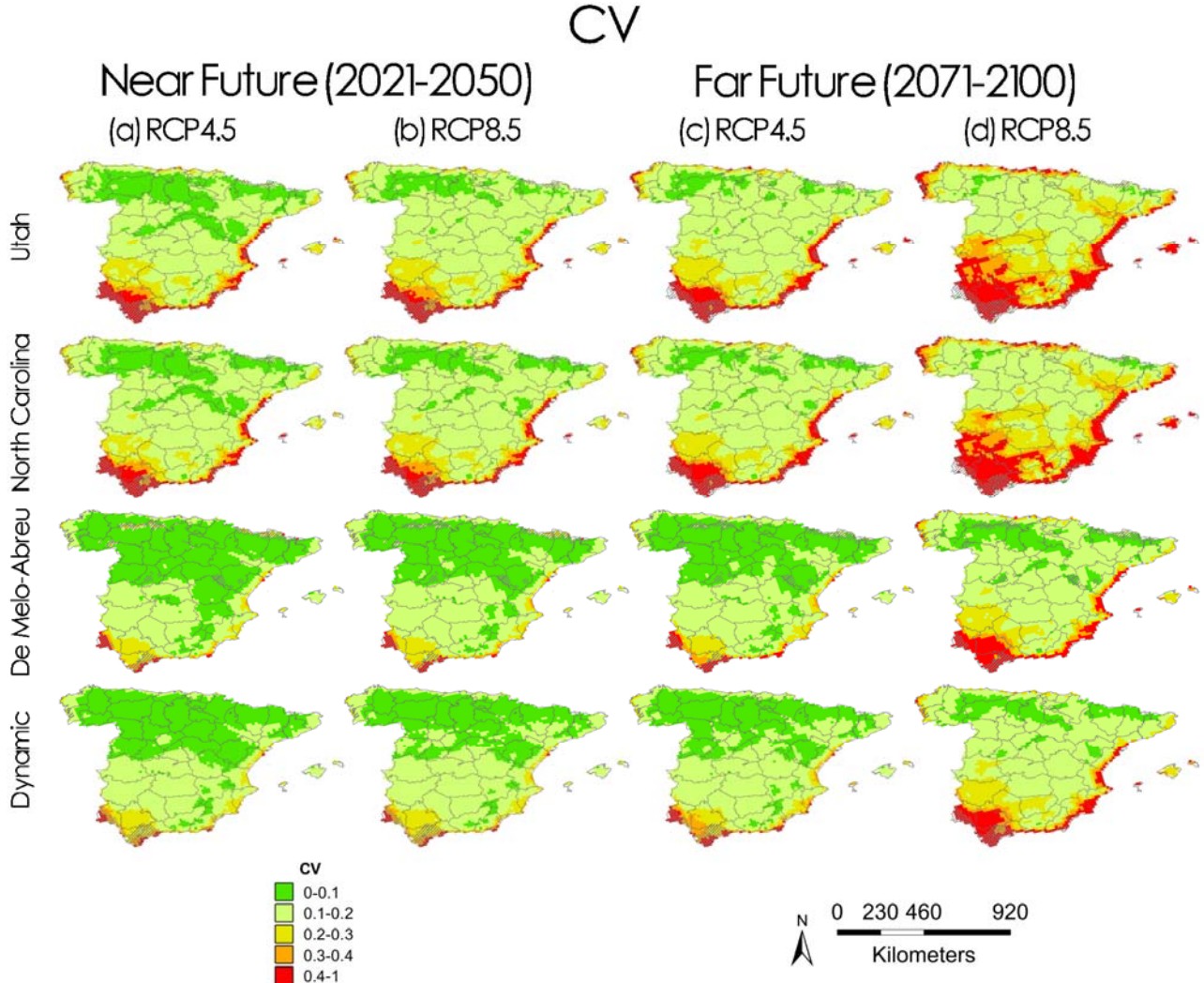

**Figure 7. Maps of inter-annual variability results calculated with the 10 EUR-11 ensemble members' mean coefficient of variation (CV, expressed per unit) of the 30-year period. For each chilling model (rows), for RCP4.5 and RCP8.5 scenarios in the 2021-2050 period (first and second column respectively) and in the 2071-2100 period (third and fourth column respectively). Grid cells with mean absolute percentage error (MAPE, %) values from the validation phase higher than 20 are not considered as reliable results and are highlighted in diagonal lines.**

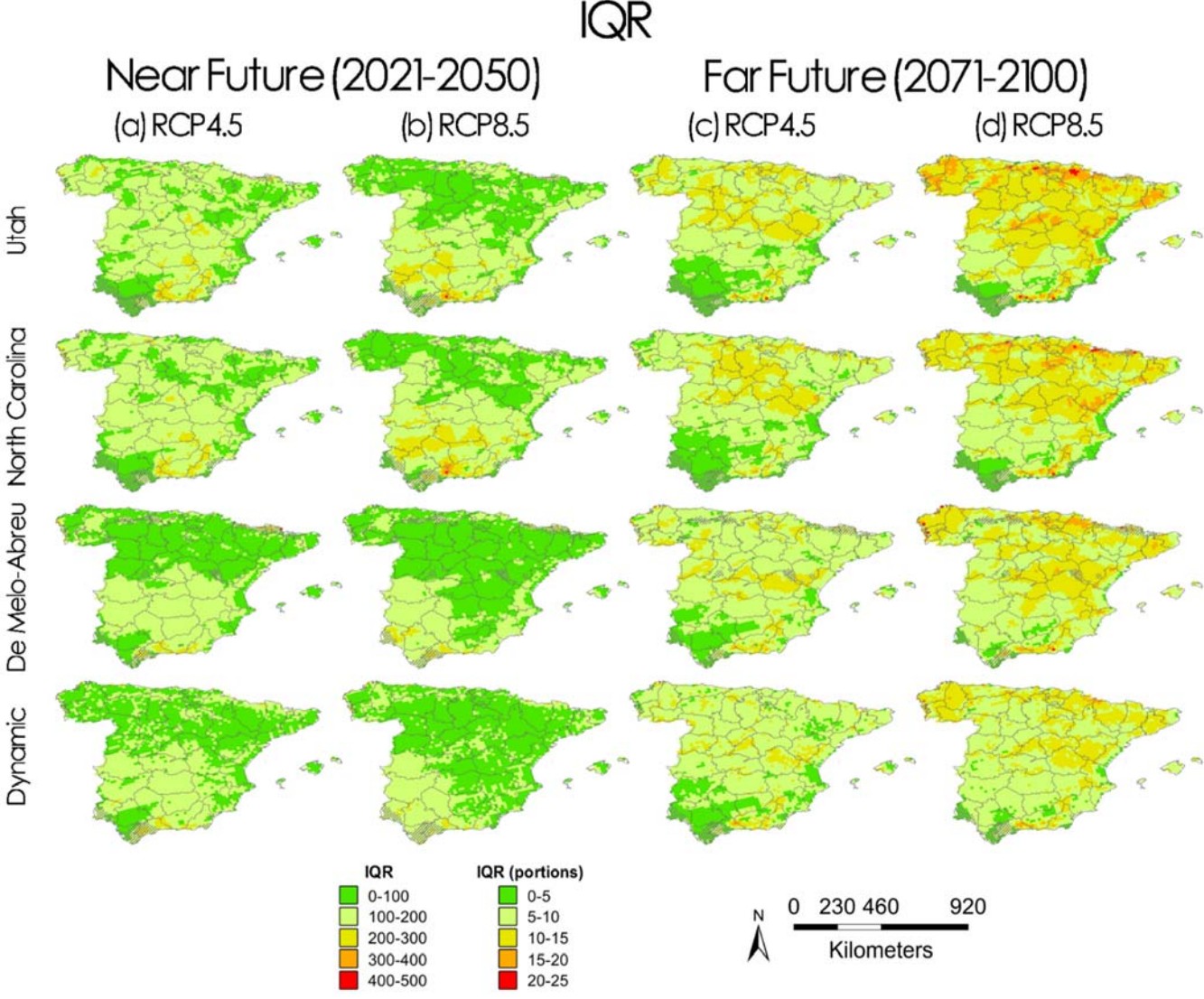

**Figure 8. Maps of uncertainty results calculated with the ensemble's interquartile range (IQR) of the 30-year mean chilling sums of the 10 EUR-11 ensemble members. For each chilling model (rows), for RCP4.5 and RCP8.5 scenarios in the 2021-2050 period (first and second column respectively) and in the 2071-2100 period (third and fourth column respectively). Grid cells with mean absolute percentage error (MAPE, %) values from the validation phase higher than 20 are not considered as reliable results and are highlighted in diagonal lines.**

**Table 1. Chilling requirements gathered from the existing literature for some of the main varieties of apple, olive and peach trees, using different chilling calculation models: North Carolina (chilling units), Dynamic (chilling portions), Utah (chilling units) and De Melo-Abreu (chilling units). Values were rounded to the nearest integer value. Mean value was calculated when values for the same variety were found in more than one source. Mean number of seasons (out of 29) for the studied periods where chilling requirements are compromised at the indicated location.**

| Tree crop | Chilling model | Variety | Chill requirements (units/portions) and source(s) | Location | Mean number of compromised seasons | | | | |
|---|---|---|---|---|---|---|---|---|---|
| | | | | | 1976-2005 | 2021-2050 | | 2071-2100 | |
| | | | | | | RCP4.5 | RCP8.5 | RCP4.5 | RCP8.5 |
| Apple | North Carolina | Royal Gala | 1049 [b] | Lleida | 0 | 0.7 | 0.7 | 3.1 | 19.9 |
| | | Golden Delicious | 1050 [a] | (NE Spain (41º35'N, | 0 | 0.7 | 0.7 | 3.1 | 19.9 |
| | | Granny Smith | 1057 [a, b] | 0º41'E) | 0 | 0.7 | 0.8 | 3.1 | 20.3 |
| | | Fuji | 1077 [a] | | 0 | 0.7 | 0.8 | 3.5 | 20.6 |
| Olive | De Melo-Abreu | Arbequina | 339 [d] | Almonte (SW Spain, | 0 | 0.4 | 0.4 | 3.2 | 17.6 |
| | | Picual | 469 [d] | 37º13'N, | 0 | 1.6 | 3.3 | 6.6 | 23.6 |
| | | Hojiblanca | 494 [d] | 6º27'W) | 0 | 2 | 3.5 | 7.4 | 25.1 |
| Peach | Dynamic | Sunlite | 33 [e] | Buñol | 0 | 0 | 0 | 0.4 | 7.9 |
| | | Flavortop | 38 [c, e] | (E Spain, | 0 | 0 | 0 | 0.5 | 11.4 |
| | | Fantasia | 42 [c, e] | (39º22'N, | 0 | 0.3 | 0 | 1.9 | 15.1 |
| | | Redhaven | 73 [c, f] | 0º48'W) | 2.3 | 14.2 | 18 | 24 | 28.6 |
| Peach | Utah | Sunlite | 536 [e] | Murcia | 0.8 | 9.8 | 11.3 | 17.6 | 27.7 |
| | | Flavortop | 657 [c, e] | (SE Spain, | 2.7 | 14.6 | 16.9 | 23.7 | 28.6 |
| | | Fantasia | 683 [c, e] | (37º56'N, | 3.1 | 15.2 | 17.8 | 24.2 | 28.6 |
| | | Elberta | 790 [g] | 1º10'W) | 6.9 | 20.8 | 21.6 | 25.8 | 28.6 |
| | | Redhaven | 813 [c, g] | | 7.7 | 21.9 | 22.5 | 26.2 | 28.6 |

[a] Hauagge and Cummins (1991) [b] Cook et al. (2017) [c] Erez (2000) [d] De Melo-Abreu et al. (2004) [e] Linsley-Noakes and Allan (1994) [f] Roman et al. (1998) [g] Richardson et al. (1974)

