# Peer review of "Chilling accumulation in temperate fruit trees in Spain under climate change"

_Natural Hazards and Earth System Sciences, 2018_

## Referee Comment (RC1) · Luedeling (Referee) · 3 Feb 2019

Rodriguez et al. present an assessment of past and future winter chill in Spain, using an ensemble of climate scenarios and four chill models. It seems to me that the climate data processing was very well done; the way scenarios were prepared seems very reasonable. The authors' expertise in this field is evident. Unfortunately, the study has some shortcomings regarding the estimation of winter chill, which will have to be addressed. Major issues: 1) Similar work has been done before, for various countries and also at global scale. It remains somewhat unclear what the particular advantage of this new approach is. A (smaller) ensemble approach was already used 10 years ago (Luedeling et al., 2009a) for California and shortly afterwards at the global scale (Luedeling et al., 2011). In these studies, we used a weather generator rather than

just climate model outputs, which (in my view) makes the methodology used then more robust than what is presented here. Admittedly, some other elements of these assessments were not as well done as what is described in the current manuscript, and it's good to see a study using RCPs rather than SRES scenarios (though we did this here: Benmoussa et al., 2018, but not as a spatial analysis), but the novelty of the current methodology isn't sufficiently described. 2) Another innovation the authors point out isn't really a feature but rather a bug in my view. As highlighted on page 9, ll. 1-2, this may well be the first study that projected climate change impacts for these four chill models. However, there are good reasons for there not being more studies, in particular no recent studies. The reason is simply that most of these models can't be trusted to accurately describe chill accumulation. There have been a number of model comparisons over the years that have consistently found the Dynamic Model to be superior to the others (e.g. Benmoussa et al., 2017; Luedeling et al., 2009b; Ruiz et al., 2007; Zhang and Taylor, 2011; there are quite a few more). Adding old, obsolete models to such a study would be like adding a flat-earth model to a GCM ensemble – it makes little sense to consider models that have been shown to be inadequate. The situation with chill models is not the same as with GCMs – we do have a clear idea of which models are better, and there is really no rationale in my view to go for an ensemble approach. 3) Related to the previous points, we've done several studies to compare the response of various chill metrics to climate change. First, they differ greatly in their sensitivity to warming (Luedeling et al., 2009c). Second, they are not comparable, with the ratio between different chill metrics varying tremendously across the globe, especially along climate gradients (Luedeling and Brown, 2011). Especially at the warmest end of the climatic range for temperate fruit trees, most models fail (Balandier et al., 1993; Benmoussa et al., 2017a, 2017b; Linsley-Noakes and Allan, 1994). The Dynamic Model is the only model I know that has a chance of somewhat describing changes correctly across different climates. This is the reason why in our 2011 paper (Luedeling et al., 2011) we only report Chill Portions (we actually calculated other metrics too, if I remember correctly, but I consider the results meaningless).

This reasoning is actually described in several places in this paper and elsewhere (e.g. Luedeling, 2012). Just as an illustration, in the literature we found the chilling requirement of 'Ohadi' pistachios quantified at 1000+ CH in Turkey, but they grow well at 100 CH in Tunisia. This difference is not trivial at all and illustrates how badly off we can be if we use the wrong model. 4) One particular criticism of chill models has been that they are calibrated for a particular site and not necessarily generally valid. There is a reason why the North Carolina Model and the Utah Model are named after geographic areas, not after crops, and why researchers in various places saw the need to make adjustments. For example, in South Africa the Utah Model regularly produced negative chill totals at the end of the season. This was 'addressed' by removing the chill negation (resulting in the Positive Utah Model: Linsley-Noakes and Allan, 1994). The necessity of these 'empirical hacks' clearly indicates that these models can't be trusted across climatic gradients – which is critically important for a credible climate change assessment. 5) The presumably innovative outlook of possibly using estimates of the amount of chill that is exceeded 90% of the time (p. 10, l. 29) isn't so innovative after all. In fact, we already used this 'Safe Winter Chill' approach in several publications, dating back to 2009 (Luedeling et al., 2009a, 2011). It has also been picked up by others (though I don't currently remember who that was). 6) Another alleged innovation is the variable duration of the chilling period, which is determined by the minimum and maximum chill accumulation. Sure, this is new, but is it correct? The authors don't present any evidence for this. I realize that some authors have claimed that something like this makes sense (e.g. Cesaraccio et al., 2004 for their own model, but others have also said this for the Utah Model I think), but is there really any evidence? Actually, I strongly doubt that trees can make use of chill accumulation over the entire cold period. We've done a number of studies where we tried to statistically determine the chill-responsive period (Guo et al., 2015; Luedeling and Gassner, 2012; Luedeling et al., 2013a, 2013b), and we've always found periods that are much shorter than the full winter season. Now this may mean various things, including that trees are pretty safe from chill shortfalls in many places, but I suspect that it would make sense to end the

chilling period earlier than an automatic algorithm would suggest (actually, if I could change one thing about our earlier studies, I would shorten the period we considered, which seems much too long now in hindsight). 7) The paper starts with a strange introduction about the classification of fruit trees, which I'm not sure I agree with and which is also not relevant here. This paper is only about temperate species, so no need for such a general take. The first two paragraphs should be deleted. 8) I strongly urge the authors to make their code public, either in a repository or as supplementary materials to this paper. This will make it much easier to understand what was done. For instance, the statement that the authors used the method by Fishman et al. (1987a, 1987b) is not sufficiently detailed – anyone who's seen these papers knows that this is not at all trivial to implement (and I wonder if this is really the authors' source of the algorithm). Ideally, a paper should be reproducible, meaning that the methods should be sufficiently detailed for readers to repeat an experiment. This is often not really achievable, but it is not difficult for a modeling study such as the one described here. Please share the code. The main reason for this is that the actual results of this paper are not particularly helpful – pretty much the same has been shown before. The innovation (for the chill modeling community) lies in the climate data processing, but if this isn't actually shared with readers, nobody can easily make use of this methodology. In my view, the offer that readers can contact the authors isn't sufficient. 9) Finally, I suggest that the authors compare their results (and maybe also their methods) with similar studies that have been done before. There have been quite a few, as the authors will realize if they do a systematic search, not necessarily on Spain, but on various other regions. 10) Even more finally, I suggest language editing. There is still some room for improvement in terms of language, and some statements are unclear. Minor issues: p. 1, l. 14: what are 'inner physiological factors'? p. 1, l. 14: 'accumulating cool temperatures to finish dormancy is unclear (at least in terms of what dormancy this is – I'd most likely associate finishing dormancy with bloom of leaf out, but that also requires heat). No need for "be broken" in quotation marks. This is commonly used and doesn't need to be identified as an odd term (or whatever the purpose of the quotation

marks is). p. 1, l. 16: I don't think the chilling requirement is different for each variety (which means that no two varieties have the same requirement). They are crop and variety-specific, but not all different. p. 1, l. 28 – p.2, l. 10: irrelevant – delete p. 2, l. 12: income, not wealth p. 2, several places: for simplicity and reader-friendliness, I recommend replacing 10ˆ6 by 'million' p. 2, ll. 18-19: FAOSTAT doesn't directly provide such values I believe, so it would be important to state how this was determined (also note that there are all kinds of issues with this database). It is also not obvious that this sentence refers to the global scale, since the previous sentence talks about Spain. Overall, this isn't a very relevant statement in a paper that's just on Spain. p. 2, l. 24: I believe the thing trees are sensitive to is frost (not generally cold temperatures) p. 3, l. 1: 'accumulation of cold periods' is an unfortunate choice of words. Sounds like trees need, say, 5 cold periods to break endodormancy. p. 3, l. 3: not all models are based on temperatures between certain thresholds. The Dynamic Model works differently, and even the Utah-type models don't really follow this simple structure. p. 3, 12: I disagree that the chilling requirement corresponds to conditions where a tree is grown. It may rather correspond to conditions where it evolved/was bred p. 3, ll. 13-17: not sure what information is conveyed here. The initial statement is about considering a range, but then the examples are precise values, not ranges. If this is supposed to illustrate intra-specific variation, then please make sure to use the appropriate terminology (not sure what 'crop tree' refers to). p. 4, l. 9 (or elsewhere): Somewhere the authors need to mention the various chill assessments that have already been done by a number of people in a wide range of places. p. 4, l. 17: no, the models do not need hourly Tmin and Tmax. They just need hourly temperature, which can be derived (if no other information available) from daily Tmin and Tmax. p. 4, l. 22: not sure what 'freely distributed' means. Open-access? p. 4, l. 24: is this really an observational dataset? p. 5, l. 15: more details are needed on the temperature generation, especially since the source will be hard to find for most readers. What mathematical functions were used for constructing daily curves? The common method in horticultural studies such as this one is a methodology by Linvill (1990), which is based on a sine curve during

the day and logarithmic cooling at night (implemented in the chillR package; Luedeling, 2018). I'd be quite curious to learn how de Wit's method compares with this, but the authors provide insufficient information about their approach. p. 6, ll. 11-13: The authors compute a mean and then a median. Later in the paper they argue that one should calculate a 10% quantile. Why didn't they do this here? p. 6, ll. 16-17: As stated above, I'd prefer to have the code made publically available, for full transparency and usefulness. p. 6, l. 23: Is the full name of MAPE really 'mean percentage absolute error'? That would seem to lead to the acronym 'MPAE' p. 7, l. 19: 'similarly simulated' is awkward wording p. 7, ll. 23-27: All these models use different units, so they can't be compared (the fact that they're probably all called chill units doesn't make them equivalent). While it's obvious that the Dynamic Model can't be compared to the others (because values are much smaller than for the other metrics), the others are also not comparable! p. 8, l. 27: scenarios were averaged in this study, but we also provided information for determining the impact of climate model and emissions scenario. p. 9, ll. 1-2: As stated above, I don't consider it an asset to include outdated models in a study... p. 9, l. 22: not sure what 'discrete nature' means. And I also think that this may be an indication that these models are too sensitive for warm places. p. 9, ll. 26: this study didn't 'find' this, it just reported on it. Luedeling et al. (2011) sort of found this. p. 10, ll. 4: Yes, it would be great to have more datasets, but we actually already have a lot. Rather than call for collecting more data, I'd call for better use of such data for model development and validation. p. 10, ll. 11-12: 'crop varieties depending on the RCP' is unfortunate wording. First, crop varieties don't depend on RCP. Second, RCPs are theoretical pathways that will not be followed precisely. Better to say something like 'depending on how rapidly GHG emissions can be reduced' or something like that. p. 10, l. 23: not sure what 'low-limit chill requirements' are p. 10, l. 29: as mentioned above, this is exactly what the Safe Winter Chill metric achieves. p. 11, 2-4: It's obvious that RCP8.5 causes greater change, similar to the end vs. middle of century. Doesn't need to be mentioned or should clearly be marked as expected. p. 11, l. 6: why especially in warm regions? The impact depends not only on chill loss, but also on

what is grown there and how much chill it needs. p. 11, ll. 17-18: confusing sentence. Reference list: It would be so much easier to look through this, if all but the first row of a reference were indented. Maps: maps should have a coordinate system, north arrow, scale bar etc. Fig. 1: I doubt that all the olive data are right. If so, some parts of Spain would be almost exclusively olives. Maps 3-7: very hard to compare changes, which is really the most important part of this paper, if the maps are scattered across various places. Fig. 5: is the scale used for the change useful.

In summary, I think this contribution has potential, since the way the climate data were processed is very robust. But the team should consider adding some chill modeling capacity to the study to make this more convincing. While chill seems like an easy application of a climate change projection framework (it's assumed to just depend on temperature after all), things are actually quite complicated due to the invisibility of chill-induced changes, which has precluded the development of convincing models so far. In consequence, there are many models, and most of them are not suitable for studies across climates. If the authors manage to adequately consider this, this manuscript may become publishable.

References: Balandier, P., Bonhomme, M., Rageau, R., Capitan, F., and Parisot, E. (1993). Leaf bud endodormancy release in peach trees - evaluation of temperature models in temperate and tropical climates. Agricultural and Forest Meteorology 67, 95–113. Benmoussa, H., Ghrab, M., Ben Mimoun, M., and Luedeling, E. (2017a). Chilling and heat requirements for local and foreign almond ( Prunus dulcis Mill.) cultivars in a warm Mediterranean location based on 30 years of phenology records. Agricultural and Forest Meteorology 239, 34–46. Benmoussa, H., Luedeling, E., Ghrab, M., Ben Yahmed, J., and Ben Mimoun, M. (2017b). Performance of pistachio ( Pistacia vera L.) in warming Mediterranean orchards. Environmental and Experimental Botany 140, 76–85. Benmoussa, H., Ben Mimoun, M., Ghrab, M., and Luedeling, E. (2018). Climate change threatens central Tunisian nut orchards. International Journal of Biometeorology 62, 2245–2255. Cesaraccio, C., Spano, D., Snyder, R.L., and Duce, P. (2004).

Chilling and forcing model to predict bud-burst of crop and forest species. Agricultural and Forest Meteorology 126, 1–13. Fishman, S., Erez, A., and Couvillon, G.A. (1987a). The temperature dependence of dormancy breaking in plants - computer simulation of processes studied under controlled temperatures. Journal of Theoretical Biology 126, 309–321. Fishman, S., Erez, A., and Couvillon, G.A. (1987b). The temperature dependence of dormancy breaking in plants - mathematical analysis of a two-step model involving a cooperative transition. Journal of Theoretical Biology 124, 473–483. Guo, L., Dai, J., Wang, M., Xu, J., and Luedeling, E. (2015). Responses of spring phenology in temperate zone trees to climate warming: A case study of apricot flowering in China. Agricultural and Forest Meteorology 201, 1–7. Linsley-Noakes, G.C., and Allan, P. (1994). Comparison of 2 models for the prediction of rest completion in peaches. Scientia Horticulturae 59, 107–113. Linvill, D.E. (1990). Calculating chilling hours and chill units from daily maximum and minimum temperature observations. HortScience 25, 14–16. Luedeling, E. (2012). Climate change impacts on winter chill for temperate fruit and nut production: A review. Scientia Horticulturae 144, 218–229. Luedeling, E. (2018). chillR: Statistical methods for phenology analysis in temperate fruit trees (R package). Luedeling, E., and Brown, P.H. (2011). A global analysis of the comparability of winter chill models for fruit and nut trees. International Journal of Biometeorology 55, 411–421. Luedeling, E., and Gassner, A. (2012). Partial Least Squares Regression for analyzing walnut phenology in California. Agricultural and Forest Meteorology 158–159, 43–52. Luedeling, E., Zhang, M., and Girvetz, E.H. (2009a). Climatic Changes Lead to Declining Winter Chill for Fruit and Nut Trees in California during 1950–2099. PLoS ONE 4, e6166. Luedeling, E., Zhang, M., McGranahan, G., and Leslie, C. (2009b). Validation of winter chill models using historic records of walnut phenology. Agricultural and Forest Meteorology 149, 1854–1864. Luedeling, E., Zhang, M., Luedeling, V., and Girvetz, E.H. (2009c). Sensitivity of winter chill models for fruit and nut trees to climate change. Agriculture, Ecosystems & Environment 133, 23–31. Luedeling, E., Girvetz, E.H., Semenov, M.A., and Brown, P.H. (2011). Climate Change Affects Winter Chill for Temperate Fruit and Nut Trees.

[Figure]

PLoS ONE 6, e20155. Luedeling, E., Guo, L., Dai, J., Leslie, C., and Blanke, M.M. (2013a). Differential responses of trees to temperature variation during the chilling and forcing phases. Agricultural and Forest Meteorology 181, 33–42. Luedeling, E., Kunz, A., and Blanke, M.M. (2013b). Identification of chilling and heat requirements of cherry trees—a statistical approach. International Journal of Biometeorology 57, 679–689. Ruiz, D., Campoy, J.A., and Egea, J. (2007). Chilling and heat requirements of apricot cultivars for flowering. Environmental and Experimental Botany 61, 254–263. Zhang, J., and Taylor, C. (2011). The Dynamic Model provides the best description of the chill process on "Sirora" pistachio trees in Australia. HortScience 46, 420–425.

---

## Referee Comment (RC2) · Bianca Drepper (Referee) · 8 Feb 2019

Alfredo Rodríguez et al. did an extensive and rigorous job on trying to quantify future developments on chilling accumulations for Peninsular Spain and the Balearic islands. They did a major effort in modelling and validation of input data and consider a highly relevant aspect of local fruit production that is vulnerable to climate change (Campoy et al., 2011; Luedeling, 2012). In this sense, and in my opinion, this regional study has its relevance and its place in this journal. This study does also contribute to a better understanding in this domain, by improving the methodology with regards to previous studies through the use of state of the art climate models and scenarios, although it does not stand out for the novelty of the used approaches. To increase the value, that the paper brings to the scientific community as well as to end users, a couple of

revisions are suggested below, which, if taken into account, would make this paper more suitable for publication.

Major remarks regarding the content

With regards to the methodology and scope of the paper, I agree in most points with Eike Luedelings review comment (RC1):

(1) First of all, combining models that have been found to be inadequate (Luedeling, 2012) is not innovative, and the fact that the models were apparently applied without calibration to local conditions is in my eyes the biggest shortcoming of the paper. To my knowledge, there is no evidence that a model that was tested for North Carolina (Latitude range ∼36.5°N-33.8°N, Köppen-Geiger classification 'Warm temperate with hot summer climate' (Peel et al., 2007)), can be transferred to Spain (Latitude range ∼43.5, 36.0, major Köppen-Geiger classification 'Arid steppe cold' climate (Peel et al., 2007)); nor can be safely assumed, that the cultivars in all regions have the same physiology, which is implied by using the same model, despite the mention of this fact on p.3, l. 11.

(2) At this point of the introduction, a better contextualization and reference for the obtained values would be highly appreciated. Only on p.3, ll. 14-17, an exemplary chilling requirement is given, and this for apricot which is not considered in this study. Without a knowledge of local requirements of apple, olive and vineyard, the severity of the change in chilling units is hard to grasp. Also, with the quoted requirements at hand ("631 chill units [Utah model, 'Palsteyn' variety), the observed difference between models ("less than 500 chill units", p.7 l. 25) can be substantial, and the outcome of Figures 7-8 more alarming than described in the paper. Later, on p.10, ll. 11- 17, exemplary requirements for an apple and an olive variety is given, which are at risk of not being fulfilled according to the 'far future' predictions. For better understanding of the key findings of the paper, more such values should be given.

(3) In my opinion, estimations of concrete, crop or variety related shortcomings in chilling have highest relevance for planning applications and various end users, so if this is possible, it would be very interesting to find in this paper indications which zones under cultivation of a given crop will become unsuitable in terms of chilling for major varieties.

(4) Obviously, the diversity of species cannot be fully covered in this paper, but, joining the suggestion of RC1, with open source code and output maps, interested parties could quickly assess these zones following an example. It might be a subject of discussion in this stage of the paper, if these findings would be improved or not by considering the agreement of different chilling models. A priori, there is a major concern with this methodology, that I share with the author of RC1, because of the unjustified comparison of chill units among models and the mentioned inadequacy of some of them.

(5) Potentially reducing the number of models and increasing the documentation (equations, parameters) of the models should help overcome the, in my view, given uncertainty about how the different models can be understood considering the three studied fruit crops mentioned in the paper. In p.5 ll.22-24, the North Carolina model is introduced as being developed for apple trees, the De Melo-Abreu method for olive trees and the Dynamic method for peach trees. I would wish for more elaboration on how these choices have been justified and on how to make use of the findings presented in the Figures 3-8. The codes should be open access, too, since I totally agree with RC1, a research should be reproducible and with the given information this is not of application.

Remarks regarding the form

Title

In p.2, ll.1-22, the authors state "Vineyard, apricot trees, olive trees and almond trees could be also included in this last subgroup [of temperate fruits], although some of their climatic requirements are nearer the subtropical fruit trees" p.2,ll 6-7). Bearing this in mind, the mention 'temperate fruit trees' in the title of the paper is in my opinion a bit misleading, although reference handbooks do classify olives and grape as temperate

(Schaffer, 2018).

Abstract

The abstract could be more concise and feature more detail about the findings of this study than the context.

Introduction

In p.2, ll.1-22: In line with RC1, I consider the description of the classification as too long and can be left out, especially in view of the ambiguity of the classification mentioned above. The section on bias adjustment (p.4 ll.1-9) could be slightly more elaborated, and precise how it is ensured that the change over time of the climate signal is not cancelled out, see also Michelangeli et al. (2009). The transition from this paragraph to the following is a bit sharp. At this point, an overview of similar (regional) studies on chilling requirements would be expected point.

Materials and methods

Regarding the selection of models and scenarios, although hardly done in literature, the choice of models could be better justified using methodologies as in (Mendlik and Gobiet, 2016), since there is evidence of high sensitivity of climate model selection (Wilcke and Bärring, 2016). However, the authors chose the two reasonable scenarios (RCP 4.5 and RCP 8.5), allowing for consistent comments on importance of mitigation in context of actual discussion. Key equations of the chilling models should be provided in the additional material. In the main text a comment on the validation of the models should be given, in the view of their applicability on future time series.

Results

With regards to the CV, MAPE and IQR, the classes > 20, >0.4... are in my view not informative enough. Also, in section 3.1, the MAPE values are declared as problematic above 20% for few grid points, without mentioning until how high they stretch. Thus, no conclusion can be made if the computation for these grid points can be trusted at all.

[Figure]

Discussion

The difference between the two researched scenarios could be expressed more clearly (p.10, ll.11-19).

References

I join the request made in RC1 for indented references. In the text, the reference in p.3, l.29 should be revised.

Figures

As stated in RC1, all figures need to be presented with a scale bar, north arrow, and (due to inconsistency between figures) the reference system. Preferably all maps would be shown in the same projection (or the stretch of the figures should be revised). The layout of subfigures could be optimized so as to allow for bigger figures. If the decision will be taken to not report on all models, this could be of great improvement of the readability.

Figure 1 shows a good overview of land use in Spain for the reader, exposing major growing areas for the considered crops. Values seem reasonable from my experience. However, the choice of the color map is unfortunate, <1%, which could be conceptually be negligible, is very hard to distinguish from the higher classes. I suggest to revise the classification to a lower number of classes, 5 being preferred. A clarification is needed whether the map shows the percentage from the total area or from area classified as cropland.

Figure 2 features a useful example output of the analysis, but it was not justified that this is a representative example. The most reliable model would have been preferred, the Dynamic model was judged as best performing (Luedeling, 2012). In subplot B, over the years, the chilling units decrease, a trend line could be interesting, next to the mean. Subplot C should highlight which model is used for subplots A and B. In subplot D, neighboring grid points expose substantial differences in this mountainous terrain.

With regards to the shortcomings mentioned in mountains areas, a further study could envision a more focused analysis on those areas.

Regarding Figures 3 -8 and as mentioned above, classes such as >20 are little informative. In this line, it would be of great value if the maps could either exclude or highlight less reliable outcomes. This could be done by keeping grid points white, or, if readability is not compromised, with a hatched overlay. From visual comparison, there seems to be a substantial part of the apple cultivation shown in Figure 1 in coastal and mountainous areas, those reported as with comparatively high errors.

Technical comments (additionally to those mentioned in RC1, to which I fully agree):

* P.2 l. 26 delete 'it

* P.2 l.18, production, not productivity (if productivity is meant, the reference i.e. area should be specified, and I agree it is not relevant in this paper, rather give the importance of other fruits in Spain, ideally with national statistics rather than FAOSTAT)

* P.3 l.34, add 'among other regions'

* P.5 l.23 inconsistent usage of Dynamic model / Dynamic method

* P.8 l.10, specify where the biggest change occurred

* P.9 l.16, Mediterranean','

* P.9 ll.17-18 reformulate

* P.9.l.21 a warmer scenario

* P.9 ll.28-29 'Nonetheless, few tree crops are grown [. . .]' – have these areas also be found as potential new cropping areas?

* P.10 l.17 are you comparing this value (469 chilling units, according to the De Melo-Abreu method) with all outputs? It should only be compared to the output of the analysis using the same method, which, in the case of the far future under RCP8.5, where

the map shows mainly values between 500-1000 chill units in the area coinciding with olive cropping.

References

Campoy, J.A., Ruiz, D., Egea, J., 2011. Dormancy in temperate fruit trees in a global warming context: A review. Sci. Hortic. 130, 357–372. https://doi.org/10.1016/j.scienta.2011.07.011 Luedeling, E., 2012. Climate change impacts on winter chill for temperate fruit and nut production: A review. Sci. Hortic. 144, 218–229. https://doi.org/10.1016/j.scienta.2012.07.011 Mendlik, T., Gobiet, A., 2016. Selecting climate simulations for impact studies based on multivariate patterns of climate change. Clim. Change 135, 381–393. https://doi.org/10.1007/s10584-015-1582-0 Michelangeli, P.-A., Vrac, M., Loukos, H., 2009. Probabilistic downscaling approaches: Application to wind cumulative distribution functions. Geophys. Res. Lett. 36. https://doi.org/10.1029/2009GL038401 Peel, M.C., Finlayson, B.L., McMahon, T.A., 2007. Updated world map of the KoÂĺppen-Geiger climate classification. Hydrol Earth Syst Sci 12. Schaffer, B., 2018. Handbook of Environmental Physiology of Fruit Crops: Volume I: Temperate Crops, 1st ed. CRC Press. https://doi.org/10.1201/9780203719299 Wilcke, R.A.I., Bärring, L., 2016. Selecting regional climate scenarios for impact modelling studies. Environ. Model. Softw. 78, 191–201. https://doi.org/10.1016/j.envsoft.2016.01.002
* * *

---

## Author Comment (AC2) · 4 Mar 2019

*We thank the reviewer for her thoughtful comments. Our answers are highlighted in green italics.*

Alfredo Rodríguez et al. did an extensive and rigorous job on trying to quantify future developments on chilling accumulations for Peninsular Spain and the Balearic islands. They did a major effort in modelling and validation of input data and consider a highly relevant aspect of local fruit production that is vulnerable to climate change (Campoy et al., 2011; Luedeling, 2012). In this sense, and in my opinion, this regional study has its relevance and its place in this journal. This study does also contribute to a better understanding in this domain, by improving the methodology with regards to previous studies through the use of state of the art climate models and scenarios, although it does not stand out for the novelty of the used approaches. To increase the value, that the paper brings to the scientific community as well as to end users, a couple of revisions are suggested below, which, if taken into account, would make this paper more suitable for publication.

*We appreciate the referee's comments.*

**Major remarks regarding the content**

With regards to the methodology and scope of the paper, I agree in most points with Eike Luedelings review comment (RC1):

**(1)** First of all, combining models that have been found to be inadequate (Luedeling, 2012) is not innovative, and the fact that the models were apparently applied without calibration to local conditions is in my eyes the biggest shortcoming of the paper. To my knowledge, there is no evidence that a model that was tested for North Carolina (Latitude range 36.5N-33.8N, Köppen-Geiger classification 'Warm temperate with hot summer climate' (Peel et al., 2007)), can be transferred to Spain (Latitude range 43.5, 36.0, major Köppen-Geiger classification 'Arid steppe cold' climate (Peel et al., 2007)); nor can be safely assumed, that the cultivars in all regions have the same physiology, which is implied by using the same model, despite the mention of this fact on p.3, l. 11.

- *About the comment on combining models, please see Answer#2 to referee 1*

- *About the comment on applying models to other countries without additional adjustment, please see Answer# 4 (and also 2 and to 3) to referee 1, and the references included there.*

*Additionally, we have checked the locations used to develop North Carolina model (Shaltout and Unrath, 1983). The paper says "A chill unit model was developed for 'Starkrimson Delicious' (Malus domestica Borkh.) apples grown under the **wide range of temperature and elevations in North Carolina**", and the locations considered were Wake, Cleveland, Wilkes, Mitchell, Henderson. Looking at the climate at these locations we can see that parts of Spain (northern Spain) actually share the temperature regime with them, which can be checked by looking at the second subindex of the Köppen-Geiger climate classification (a or b, denoting temperature regime, "hot summer" and "warm summer" respectively; see images below from [http://koeppen-geiger.vu-wien.ac.at/pdf/kottek_et_al_2006_A4.pdf](http://koeppen-geiger.vu-wien.ac.at/pdf/kottek_et_al_2006_A4.pdf), and Peel et al., 2007), which are the same for North Carolina and the northern Spain.*

[Figure]

*Source: [http://koeppen-geiger.vu-wien.ac.at/pdf/kottek_et_al_2006_A4.pdf](http://koeppen-geiger.vu-wien.ac.at/pdf/kottek_et_al_2006_A4.pdf)*

[Figure]

*Source: Peel et al., (2007)*

*That is why in answer#4 to referee 1 we proposed: In our case, the main driver is temperature regime; and actually, in the case of North Carolina model for*

*apples, main production area is North Spain, with climatological characteristics (temperature) more similar to North Carolina than the Spanish average. Accordingly, we will delimit more the concrete area of the apple tree production in the introduction section.*

*Peel, M. C., Finlayson, B. L., and McMahon, T. A.: Updated world map of the Köppen-Geiger climate classification, Hydrol. Earth Syst. Sci., 11, 1633-1644, 10.5194/hess-11-1633-2007, 2007.*

*Shaltout, A. D., and Unrath, C. R.: Rest completion prediction model for 'Starkrimson Delicious' apples, J. Amer. Soc. Hort. Sci., 108, 957-961, 1983.*

**(2)** At this point of the introduction, a better contextualization and reference for the obtained values would be highly appreciated. Only on p.3, ll. 14-17, an exemplary chilling requirement is given, and this for apricot which is not considered in this study. Without a knowledge of local requirements of apple, olive and vineyard, the severity of the change in chilling units is hard to grasp. Also, with the quoted requirements at hand ("631 chill units [Utah model, 'Palsteyn' variety), the observed difference between models ("less than 500 chill units", p.7 l. 25) can be substantial, and the outcome of Figures 7-8 more alarming than described in the paper. Later, on p.10, ll. 11- 17, exemplary requirements for an apple and an olive variety is given, which are at risk of not being fulfilled according to the 'far future' predictions. For better understanding of the key findings of the paper, more such values should be given.

- *To follow the referee's comment, we will add a table with values for different species showing the range of chilling requirement exhibited by the main varieties. Also, we will remove any comparative comments between methods, following referee's 1 indications.*

- *About the message from Figure 7-8: Even if the impacts are high as pointed by the referee, we wanted to stress that the wide range of chilling requirements exhibited by the varieties of a given species will facilitate adaptation. In most locations, variety change will be enough, and crop change will not be required. We will modify the text to clarify this, using the new table to illustrate it.*

**(3)** In my opinion, estimations of concrete, crop or variety related shortcomings in chilling have highest relevance for planning applications and various end users, so if this is possible, it would be very interesting to find in this paper indications which zones under cultivation of a given crop will become unsuitable in terms of chilling for major varieties.

*We will add some examples in the discussion for the highest vulnerable varieties or those with strongest market influence. We think that going into a deep analysis is beyond the scope of this paper because 1) We would need to set thresholds. As*

*illustrated by the new table, for a given crop and even just considering the most important varieties we should consider very different thresholds to estimate suitability area; 2) we would need to consider other measurements than the mean, (as the Safe Winter by Luedeling et al. (2009) and use a confidence/robustness index that allow to support any recommendation or conclusion on suitability area.*

*Luedeling, E., Zhang, M., and Girvetz, E. H.: Climatic Changes Lead to Declining Winter Chill for Fruit and Nut Trees in California during 1950–2099, PLOS ONE, 4, e6166, 10.1371/journal.pone.0006166, 2009.*

**(4)** Obviously, the diversity of species cannot be fully covered in this paper, but, joining the suggestion of RC1, with open source code and output maps, interested parties could quickly assess these zones following an example. It might be a subject of discussion in this stage of the paper, if these findings would be improved or not by considering the agreement of different chilling models. A priori, there is a major concern with this methodology, that I share with the author of RC1, because of the unjustified comparison of chill units among models and the mentioned inadequacy of some of them.

- *About including open source code in supplementary material, we have agreed to do so, see Answer#8 to referee 1.*
- *About the comment on combining or comparing models, please see Answer#2 to referee 1. In further work, we propose a methodology to assess robustness of the individual model outcomes (as they cannot be put together due to have different units), the EOA index (Rodríguez et al., 2019). Then, the agreement between chill models on the suitable zones for a given variety (as suggested by this referee in her question 2) could be compared by using the EOA values. This development, however, is out of the scope of this paper. We will clarify this in the text.*

*Rodríguez, A., Ruiz-Ramos, M., Palosuo, T., Carter, T. R., Fronzek, S., Lorite, I. J., Ferrise, R., Pirttioja, N., Bindi, M., Baranowski, P., Buis, S., Cammarano, D., Chen, Y., Dumont, B., Ewert, F., Gaiser, T., Hlavinka, P., Hoffmann, H., Höhn, J. G., Jurecka, F., Kersebaum, K. C., Krzyszczak, J., Lana, M., Mechiche-Alami, A., Minet, J., Montesino, M., Nendel, C., Porter, J. R., Ruget, F., Semenov, M. A., Steinmetz, Z., Stratonovitch, P., Supit, I., Tao, F., Trnka, M., de Wit, A., and Rötter, R. P.: Implications of crop model ensemble size and composition for estimates of adaptation effects and agreement of recommendations, Agric. For. Meteorol., 264, 351-362, https://doi.org/10.1016/j.agrformet.2018.09.018, 2019.*

**(5)** Potentially reducing the number of models and increasing the documentation (equations, parameters) of the models should help overcome the, in my view, given uncertainty about how the different models can be understood considering the three studied fruit crops mentioned in the paper. In p.5 ll.22-24, the North Carolina model is introduced as being developed for apple trees, the De Melo-Abreu method for olive trees and the Dynamic method for peach trees. I would wish for more elaboration on how these choices have been justified and on how to make use of the findings presented in the Figures 3-8. The codes should be open access, too, since I totally agree with RC1, a research should be reproducible and with the given information this is not of application.

- *About the models used in the study and dealing with uncertainty, please see Answer#2 and 3 to referee 1. Also, a brief history of each model origin, development and applications (references suggested in the Answers#2 and 3 will be included).*
- *About including open source code in supplementary material, we have agreed to do so, see Answer#8 to referee 1.*

**Remarks regarding the form**

**Title**

In p.2, ll.1-22, the authors state "Vineyard, apricot trees, olive trees and almond trees could be also included in this last subgroup [of temperate fruits], although some of their climatic requirements are nearer the subtropical fruit trees" p.2,ll 6-7). Bearing this in mind, the mention 'temperate fruit trees' in the title of the paper is in my opinion a bit misleading, although reference handbooks do classify olives and grape as temperate (Schaffer, 2018).

*We will remove the word "temperate" from the title, as follows:*

*"Chilling accumulation in fruit trees in Spain under climate change"*

**Abstract**

The abstract could be more concise and feature more detail about the findings of this study than the context.

*We will modify it making it more focused on results.*

**Introduction**

In p.2, ll.1-22: In line with RC1, I consider the description of the classification as too long and can be left out, especially in view of the ambiguity of the classification mentioned above. The section on bias adjustment (p.4 ll.1-9) could be slightly more elaborated, and precise how it is ensured that the change over time of the climate

signal is not cancelled out, see also Michelangeli et al. (2009). The transition from this paragraph to the following is a bit sharp. At this point, an overview of similar (regional) studies on chilling requirements would be expected point.

*We will remove the classification part. The bias adjustment section will be extended with some details on the methodology, according to the following guidelines:*

*Bias adjustment is based on a transfer function such that the marginal cumulative distribution function of the adjusted variable matches that of the observations. A complete discussion of the technique, including validation and effect on climate indices can be found in Piani et et al. (2010), Piani et al, (2010b), Dosio and Paruolo (2011), and Dosio et al. (2012), Ruiz-Ramos et al., (2016), Dosio and Fischer, (2018). Dosio (2016) showed that bias-adjustment largely improves the value of present and future threshold-based indices (e.g., the number of frost days): these indices are generally poorly simulated over the present climate, such that the projected climate change may not be reliable.*

*We will better link with next paragraph. Also, we will include the references of previous studies (see references included in the Answers to referee 1). Also, we can mention climate change differences when using bias correction methods and cancellation of climate change signal over time (Michelangeli et al, 2009; Casanueva et al., 2018).*

*Casanueva, A., Bedia, J., Herrera, S., Fernández, J., and Gutiérrez, J. M.: Direct and component-wise bias correction of multi-variate climate indices: the percentile adjustment function diagnostic tool, Clim. Change, 147, 411-425, 10.1007/s10584-018-2167-5, 2018.*

*Dosio, A., and Paruolo, P.: Bias correction of the ENSEMBLES high-resolution climate change projections for use by impact models: Evaluation on the present climate, Journal of Geophysical Research Atmospheres, 116, 10.1029/2011JD015934, 2011.https://agupubs.onlinelibrary.wiley.com/doi/full/10.1029/2011JD015934*

*Dosio, A., Paruolo, P., and Rojas, R.: Bias correction of the ENSEMBLES high resolution climate change projections for use by impact models: Analysis of the climate change signal, Journal of Geophysical Research Atmospheres, 117, 10.1029/2012JD017968, 2012.*

*Dosio, A.: Projections of climate change indices of temperature and precipitation from an ensemble of bias-adjusted high-resolution EURO-CORDEX regional climate models, Journal of Geophysical Research: Atmospheres, 121, 5488-5511, doi:10.1002/2015JD024411, 2016.*

*Dosio, A., and Fischer, E. M.: Will Half a Degree Make a Difference? Robust Projections of Indices of Mean and Extreme Climate in Europe Under 1.5°C, 2°C, and 3°C Global Warming, Geophys. Res. Lett., 45, 935-944, 10.1002/2017GL076222, 2018.*

*Michelangeli, P. A., Vrac, M., and Loukos, H.: Probabilistic downscaling approaches: Application to wind cumulative distribution functions, Geophys. Res. Lett., 36, 10.1029/2009GL038401, 2009.*

*Piani, C., Haerter, J. O., and Coppola, E.: Statistical bias correction for daily precipitation in regional climate models over Europe, Theoretical and Applied Climatology, 99, 187-192, 10.1007/s00704-009-0134-9, 2010.*

*Piani, C., Weedon, G. P., Best, M., Gomes, S. M., Viterbo, P., Hagemann, S., and Haerter, J. O.: Statistical bias correction of global simulated daily precipitation and temperature for the application of hydrological models, Journal of Hydrology, 395, 199-215, 10.1016/j.jhydrol.2010.10.024, 2010.*

*Ruiz-Ramos, M., Rodríguez, A., Dosio, A., Goodess, C. M., Harpham, C., Mínguez, M. I., and Sánchez, E.: Comparing correction methods of RCM outputs for improving crop impact projections in the Iberian Peninsula for 21st century, Clim. Change, 134, 283-297, 10.1007/s10584-015-1518-8, 2016.*

**Materials and methods**

Regarding the selection of models and scenarios, although hardly done in literature, the choice of models could be better justified using methodologies as in (Mendlik and Gobiet, 2016), since there is evidence of high sensitivity of climate model selection (Wilcke and Bärring, 2016). However, the authors chose the two reasonable scenarios (RCP 4.5 and RCP 8.5), allowing for consistent comments on importance of mitigation in context of actual discussion. Key equations of the chilling models should be provided in the additional material. In the main text a comment on the validation of the models should be given, in the view of their applicability on future time series.

*The ensemble of climate models contains 10 members, which was the whole set of models available. Additionally, Figure 3 is meant prove that our inputs are robust. To take into account the referee's concern we will include the following (bold) text in the corresponding section:*

*"..... This ensemble size is considered to be large enough by the agricultural impact community to retrieve robust results (Martre et al., 2015; Rodríguez et al., 2019).* **Due to the complex orography of the Iberian Peninsula and its remarkable climatic diversity (CLIVAR-Spain, 2010), no additional systematic selection was performed to reduce the number of RCM ensemble members (e.g., Mendlik and Gobiet, 2016). A thorough analysis in this sense would imply decomposing the Iberian Peninsula into several climatic sub-regions (Wilcke and Bärring, 2016) and would derive into a much more complex process to potentially improve already robust results.** *The outputs of the EUR-11 ensemble for two RCPs were considered: 1) +4.5 W/m2 radiative forcing increase at the end of the 21st century relative to pre-industrial levels (RCP4.5) and 2) the same but for +8.5 W/m2 (RCP8.5)."*

*CLIVAR-Spain: Climate in Spain: past, present and future. Regional climate change assessment report, Ministerio de Ciencia e Innovación (España), Ministerio de Medio Ambiente y Medio Rural y Marino (España) 978-84-614-8115-6, www.clivar.es, 2010.*

*Mendlik, T., and Gobiet, A.: Selecting climate simulations for impact studies based on multivariate patterns of climate change, Clim. Change, 135, 381-393, 10.1007/s10584-015-1582-0, 2016.*

*Wilcke, R. A. I., and Bärring, L.: Selecting regional climate scenarios for impact modelling studies, Environ. Modell. Softw., 78, 191-201, https://doi.org/10.1016/j.envsoft.2016.01.002, 2016.*

**Results**

With regards to the CV, MAPE and IQR, the classes > 20, >0.4... are in my view not informative enough. Also, in section 3.1, the MAPE values are declared as problematic above 20% for few grid points, without mentioning until how high they stretch. Thus, no conclusion can be made if the computation for these grid points can be trusted at all.

*High MAPE values can be related also to the low values of the chilling accumulation in those areas, and therefore it does not mean that necessarily the projections cannot be trusted at all. It means that we should be more careful when interpreting the results. Nevertheless, we will mark somehow the areas in the plots where values were greater than 20%. Also, we will choose a more understandable, representative classes for the figures, and the top end will be specified.*

**Discussion**

The difference between the two researched scenarios could be expressed more clearly (p.10, ll.11-19).

*We will introduce a sentence here discussing the main difference found between results obtained for each RCPs.*

**References**

I join the request made in RC1 for indented references. In the text, the reference in p.3, l.29 should be revised.

*Format change will be made, if the journal allows so.*

*The mentioned reference has already been checked and corrected.*

**Figures**

As stated in RC1, all figures need to be presented with a scale bar, north arrow, and (due to inconsistency between figures) the reference system. Preferably all maps would be shown in the same projection (or the stretch of the figures should be revised). The layout of subfigures could be optimized so as to allow for bigger figures. If the decision will be taken to not report on all models, this could be of great improvement of the readability.

- *We will modify the figures as suggested. Layout would be revised to allow the figures to be as large as possible, although this kind of composite figures are quite common in climate and impact publications (e.g. see https://www.meteo.unican.es/es/view/publications)*

- *On the models to be reported: Please see our answers above. We are convinced that our arguments are correct and sound, but if the editor and both referees ask us to remove some of the chilling models considered, we would be willing to do so.*

Figure 1 shows a good overview of land use in Spain for the reader, exposing major growing areas for the considered crops. Values seem reasonable from my experience. However, the choice of the color map is unfortunate, <1%, which could be conceptually be negligible, is very hard to distinguish from the higher classes. I suggest to revise the classification to a lower number of classes, 5 being preferred. A clarification is needed whether the map shows the percentage from the total area or from area classified as cropland.

*We will modify the figure 1 as suggested. We will specify that the percentage refers to the total area.*

Figure 2 features a useful example output of the analysis, but it was not justified that this is a representative example. The most reliable model would have been preferred, the Dynamic model was judged as best performing (Luedeling, 2012). In subplot B, over the years, the chilling units decrease, a trend line could be interesting, next to the mean. Subplot C should highlight which model is used for subplots A and B. In subplot D, neighboring grid points expose substantial differences in this mountainous terrain. With regards to the shortcomings mentioned in mountains areas, a further study could envision a more focused analysis on those areas.

*The example was considered representative for two reasons, 1) because it shows the general procedure followed for each cell to obtain an individual outcome from the climate ensembles, illustrating how the methodology aggregated yearly information and projections using different climate models, 2) to explain how the initial and final chilling accumulation dates are calculated, and this is particularly important for the Utah-based*

*methods considered in the study as warm temperatures sometimes negatively contribute to chilling accumulation. This is not the case of the Dynamic model where only positive increases of chilling accumulations are added up, being the purpose of the cited figure of illustrating one of the Utah-based methods, more complicated to explain in that sense. So Dynamic model is not as useful as the others to illustrate this.*

*The text will be changed to further explain the need of this figure and the footnote will be modified to clarify that all subplots refer to the same model.*

*We agree with the referee's suggestion that a further study more focused on mountainous areas would be very much interesting from the scientific point of view. However, our priority was to focus in main productive areas that are usually at lower regions.*

Regarding Figures 3 -8 and as mentioned above, classes such as >20 are little informative. In this line, it would be of great value if the maps could either exclude or highlight less reliable outcomes. This could be done by keeping grid points white, or, if readability is not compromised, with a hatched overlay. From visual comparison, there seems to be a substantial part of the apple cultivation shown in Figure 1 in coastal and mountainous areas, those reported as with comparatively high errors.

*As answered above, we will choose a more understandable, representative classes for the figures, and we will highlight the areas >20 in these figures to facilitate interpretation.*

Technical comments (additionally to those mentioned in RC1, to which I fully agree):

* P.2 l. 26 delete 'it

*We will delete it (unless our professional English editor suggest otherwise).*

* P.2 l.18, production, not productivity (if productivity is meant, the reference i.e. area should be specified, and I agree it is not relevant in this paper, rather give the importance of other fruits in Spain, ideally with national statistics rather than FAOSTAT)

*Yes, the referee is right, we will change it in the revised version.*

* P.3 l.34, add 'among other regions'

*The referee's suggestion will be included in the revised version.*

* P.5 l.23 inconsistent usage of Dynamic model / Dynamic method

*The referee's suggestion will be included in the revised version, using Dynamic model throughout the paper.*

\* P.8 l.10, specify where the biggest change occurred

*We will specify it in the revised version.*

\* P.9 l.16, Mediterranean','

*The referee's suggestion will be included in the revised version (unless our professional English editor suggest otherwise).*

\* P.9 ll.17-18 reformulate

*The referee's suggestion will be included in the revised version as follows:*

*"In light of the results, our hypothesis is that the stations in these areas are poorly represented by the interpolated Spain02 dataset."*

\* P.9.l.21 a warmer scenario

*The referee's suggestion will be included in the revised version.*

\* P.9 ll.28-29 'Nonetheless, few tree crops are grown [...]' – have these areas also be found as potential new cropping areas?

*Yes, at the lower part of the mountains, but the affected areas would be relatively small. That is why in our view improving the estimations for these areas would be interesting of course but not a priority.*

\* P.10 l.17 are you comparing this value (469 chilling units, according to the De Melo-Abreu method) with all outputs? It should only be compared to the output of the analysis using the same method, which, in the case of the far future under RCP8.5, where the map shows mainly values between 500-1000 chill units in the area coinciding with olive cropping.

*The comparison is established only between results from the same models. We will stress this in the revised version as it is a key point. it seems that it was not clear enough. Also, we will specify more the region we are referring to (red areas in Figure 8, first column, third row).*

*Thank you for your thoughtful revision. We have tried to address all the issues you raised. We are convinced that our arguments are correct and sound, but if the editor and both referees ask us to remove some of the chilling models considered, we would be willing to do so.*

**References**

Campoy, J.A., Ruiz, D., Egea, J., 2011. Dormancy in temperate fruit trees in a global warming context: A review. Sci. Hortic. 130, 357–372. https://doi.org/10.1016/j.scienta.2011.07.011

Luedeling, E., 2012. Climate change impacts on winter chill for temperate fruit and nut production: A review. Sci. Hortic. 144, 218–229. https://doi.org/10.1016/j.scienta.2012.07.011

Mendlik, T., Gobiet, A., 2016. Selecting climate simulations for impact studies based on multivariate patterns of climate change. Clim. Change 135, 381–393. https://doi.org/10.1007/s10584-015-1582-0

Michelangeli, P.-A., Vrac, M., Loukos, H., 2009. Probabilistic downscaling approaches: Application to wind cumulative distribution functions. Geophys. Res. Lett. 36. https://doi.org/10.1029/2009GL038401

Peel, M.C., Finlayson, B.L., McMahon, T.A., 2007. Updated world map of the Ko´lppen-Geiger climate classiïn˘A˛cation. Hydrol Earth Syst Sci 12.

Schaffer, B., 2018. Handbook of Environmental Physiology of Fruit Crops: Volume I: Temperate Crops, 1st ed. CRC Press. https://doi.org/10.1201/9780203719299

Wilcke, R.A.I., Bärring, L., 2016. Selecting regional climate scenarios for impact modelling studies. Environ. Model. Softw. 78, 191–201. https://doi.org/10.1016/j.envsoft.2016.01.002

---

## Referee Comment (RC3) · Eike Luedeling (Referee) · 12 Mar 2019

Thanks to the authors for the detailed responses to my comments. I'm not sure what the process is going forward. There doesn't seem to be a new manuscript to look at, and the format of the reply - with one long document addressing all my comments - isn't really suitable for a discussion. Can we get some editorial advice on how to move this forward?

---

## Author Comment (AC3) · 15 Mar 2019

Thank you for your active participation during the discussion phase.

The process we followed is available by this link: https://www.natural-hazards-and-earth-system-sciences.net/peer_review/interactive_review_process.html

As far as we know, we have now entered in the step number 5.

About your question regarding a new manuscript, in the description of the interactive discussion steps it can be read: "After the open discussion, the authors are expected to publish a response to all comments within four weeks, in case they have not done so during the open discussion. Based on the responses, the editor either invites the authors to submit a revised manuscript or directly rejects the manuscript."

[Figure]

We have received a message with instructions: "After your posts, you have to explicitly finalize the final-response form so that the Editor can make a decision about the further handling of your manuscript. Please note that your revised manuscript should not be prepared at this stage. When posting your author comments (ACs), you can choose between new comments or co-listing of existing ones. Please also consider replying to short comments (SCs) from the scientific community. The response to the Referees shall be structured in a clear and easy-to-follow sequence: (1) comments from Referees, (2) author's response, (3) author's changes in manuscript."

Following these instructions, we are going to update the responses document adding at each response the proposed changes to the manuscript. Then, we will submit the revised manuscript when requested by the editor.

---

## Author Response (AR1)

Guest editor report: nhess-2018-392; original title "Chilling accumulation in temperate fruit trees in Spain under climate change"

Answers are in green font.

The authors have undertaken an ambitious research in assessing winter chill across Spain, as derived from meteorological observations and climatic projections. Each of the two reviewers have provided an excellent and detailed revision of the manuscript, to which the authors have responded in a detailed manner.

The two reviewers reach consensus on a number of critical points. The interactive comments from both reviewers have been well documented and the authors have formulated solutions to take the manuscript further in two separate documents (RC1 and RC2). I agree with the solutions presented by the authors. Below I highlight some points of attention for the authors.

Thank you for your comments.

Overall the research can be documented better in the manuscript such that justice is done to the rigorous work undertaken. The processing of meteorological observations and climate scenarios, and their relation to the impact on the Spanish fruit trees uses state-of-the-art methodology. Therefore I would suggest that the authors revise the manuscript according to their documentation and replies to the reviewers.

A new version of the manuscript has been produced incorporating the answers we proposed in the previous revision phase (see revised paper attached). The documentation of the undertaken research has been improved following referee's instructions.

The following major points require the authors' attention:

1. Avoid vague descriptions and formulate more precisely what has been done, certainly in the abstract. [an example: "near and far future" is vague; define the periods "2021-2050" and "2071-2100"] Overall a focus on precise findings will improve the readability of both manuscript and abstract.

"Near and far future" terms have been removed from the paper, using "2021-2050 period" and "2071-2100 period" instead. Also, we made an effort to focus on main findings across all the manuscript and especially in the abstract and introduction, which have also been reduced following referee's advice (see revised paper attached).

2. A comprehensive review of chilling requirements for different species will be of enormous relevance and interest to an international audience. To this extent, the authors' suggestion of adding a table is excellent. References to the literature, as already documented in the authors' replies to the reviewers, could be extended to include research that is relevant to Spain or similar climatic environments (e.g. California).

New Table 1 reports the information of the chilling requirements for main varieties of different tree crops. The new references found following the referee's suggestions and

included in our reply have been incorporated in the manuscript, mainly in the discussion section, stressing those studies especially relevant for Spain or dealing with Mediterranean environments.

3. An important outcome of the research relates to winter chill reduction. It would be useful to discuss the number of times chilling requirements are compromised for the different periods studied.

Table 1 has been extended beyond we proposed in our reply to the referees to follow this suggestion from the editor (see Table 1 in the revised paper attached). We have selected some locations (from the areas mentioned in the discussion section) to compute the mean number of compromised seasons for the different periods studied, for the same varieties of the Table 1. A paragraph commenting this example has been included at the end of the results section (lines 10 to 15 in page 10 of the clean revised version) and is also mentioned in the discussion (paragraph from lines 9 to 32 in page 13 of the clean revised version).

4. The choice of keeping the different chill model results separately is underpinned by the reviewers' preference for the dynamic model, and therefore I recommend to keep the results separately as currently done. Nevertheless, a better documentation of the different chill models and temperature thresholds will clarify the comments made.

We agree with the editor's comment. Following her suggestion and those made by both referees, we have improved the documentation of the chill models used, their origin and the relationships and differences between them. This has been done mainly in section "2.2. Chilling modelling", but also in the introduction (lines 15 to 25 in page 4 of the clean revised version), discussion (from line 22 in page 11 until line 21 in page 12 of the clean revised version) and supplementary material (Codes 1S to 9S).

5. I leave it to the authors to decide whether to share their code in the supplementary material or document the formulas used.

We have included the codes in the supplementary material as it was strongly advised by the referees.

Since most of the above points have been documented in the replies to the reviewers, the revised manuscript can be reviewed by the handling editor.

In our comments we offered to send the manuscript to a professional Editing English service. The deadline set did not allow to do so (we had only 10 days to review the paper including Easter period), but we are willing to do so if the editor thinks this is needed.

**Responses to referee comment [RC1]**
*We thank the reviewer for his thoughtful comments. Our answers are highlighted in green italics.*

Rodriguez et al. present an assessment of past and future winter chill in Spain, using an ensemble of climate scenarios and four chill models. It seems to me that the climate data processing was very well done; the way scenarios were prepared seems very reasonable. The authors' expertise in this field is evident.

*Thank you for this comment*

Unfortunately, the study has some shortcomings regarding the estimation of winter chill, which will have to be addressed.

**Major issues:**

**1)** Similar work has been done before, for various countries and also at global scale. It remains somewhat unclear what the particular advantage of this new approach is. A (smaller) ensemble approach was already used 10 years ago (Luedeling et al., 2009a) for California and shortly afterwards at the global scale (Luedeling et al., 2011). In these studies, we used a weather generator rather than just climate model outputs, which (in my view) makes the methodology used then more robust than what is presented here. Admittedly, some other elements of these assessments were not as well done as what is described in the current manuscript, and it's good to see a study using RCPs rather than SRES scenarios (though we did this here: Benmoussa et al., 2018, but not as a spatial analysis), but the novelty of the current methodology isn't sufficiently described.

*We have softened the language about the novelty of our study throughout the paper and we have acknowledged as previous works the studies pointed by the reviewer (Luedeling et al. 2009, 2011). Besides, we have further described the methodology followed to design the climate ensemble, enhancing the description of the improvements and contributions of our study in the text: 1) Studies done in other countries would be of little help for Spanish farmers that previously could only find scarce information from studies performed in other regions, or worldwide with not enough resolution; 2) In recent studies working with multi-model ensembles formed by crop models, ensemble results tend to improve as the number of members of the ensemble increases, for instance, in Martre et al. (2015) the committed errors decreased as the ensemble members grow with little decrease beyond 10 members. This debate was analysed from the statistical point of view in Wallach et al. (2018). We consider that this is an improvement from the former studies using 3 climate models.*

*From this point only, in our view this work represents an improvement in terms of robustness, due to the ensemble design and composition.*

*We have clarified the text to stress that in this study we did not use the climate model outputs directly. Instead, a bias adjustment process was applied to the outputs prior to be applied to the models. The bias adjustment techniques are considered a valid alternative to apply on climate model outputs to crop models, especially suitable for handling the complex orography of the Iberian Peninsula (Maraun and Widmann, 2018). Of course, weather generators can also be a reasonable approach.*

*Luedeling, E., Zhang, M., and Girvetz, E. H.: Climatic Changes Lead to Declining Winter Chill for Fruit and Nut Trees in California during 1950–2099, PLOS ONE, 4, e6166, 10.1371/journal.pone.0006166, 2009.*

*Luedeling, E., Girvetz, E. H., Semenov, M. A., and Brown, P. H.: Climate Change Affects Winter Chill for Temperate Fruit and Nut Trees, PLOS ONE, 6, e20155, 10.1371/journal.pone.0020155, 2011.*

*Maraun, D., and Widmann, M.: Statistical Downscaling and Bias Correction for Climate Research, Cambridge University Press, Cambridge, 2018.*

*Martre, P., Wallach, D., Asseng, S., Ewert, F., Jones, J. W., Rotter, R. P., Boote, K. J., Ruane, A. C., Thorburn, P. J., Cammarano, D., Hatfield, J. L., Rosenzweig, C., Aggarwal, P. K., Angulo, C., Basso, B., Bertuzzi, P., Biernath, C., Brisson, N., Challinor, A. J., Doltra, J., Gayler, S., Goldberg, R., Grant, R. F., Heng, L., Hooker, J., Hunt, L. A., Ingwersen, J., Izaurralde, R. C., Kersebaum, K. C., Muller, C., Kumar, S. N., Nendel, C., O'Leary, G., Olesen, J. E., Osborne, T. M., Palosuo, T., Priesack, E., Ripoche, D., Semenov, M. A., Shcherbak, I., Steduto, P., Stockle, C. O., Stratonovitch, P., Streck, T., Supit, I., Tao, F. L., Travasso, M., Waha, K., White, J. W., and Wolf, J.: Multimodel ensembles of wheat growth: many models are better than one, Glob. Change Biol., 21, 911-925, 10.1111/gcb.12768, 2015.*

*Wallach, D., Martre, P., Liu, B., Asseng, S., Ewert, F., Thorburn, P. J., Ittersum, M., Aggarwal, P. K., Ahmed, M., Basso, B., Biernath, C., Cammarano, D., Challinor, A. J., De Sanctis, G., Dumont, B., Eyshi Rezaei, E., Fereres, E., Fitzgerald, G. J., Gao, Y., Garcia-Vila, M., Gayler, S., Girousse, C., Hoogenboom, G., Horan, H., Izaurralde, R. C., Jones, C. D., Kassie, B. T., Kersebaum, K. C., Klein, C., Koehler, A.-K., Maiorano, A., Minoli, S., Müller, C., Naresh Kumar, S., Nendel, C., O'Leary, G. J., Palosuo, T., Priesack, E., Ripoche, D., Rötter, R. P., Semenov, M. A., Stöckle, C., Stratonovitch, P., Streck, T., Supit, I., Tao, F., Wolf, J., and Zhang, Z.: Multimodel ensembles improve predictions of crop–environment–management interactions, Glob. Change Biol. (in press), doi:10.1111/gcb.14411, 2018.*

**2)** Another innovation the authors point out isn't really a feature but rather a bug in my view. As highlighted on page 9, ll. 1-2, this may well be the first study that projected climate change impacts for these four chill models. However, there are good reasons for there not being more studies, in particular no recent studies. The reason is simply that most of these models can't be trusted to accurately describe chill accumulation. There have been a number of model comparisons over the years that have consistently found the Dynamic Model to be superior to the others (e.g. Benmoussa et al., 2017; Luedeling et al., 2009b; Ruiz et al., 2007; Zhang and Taylor, 2011; there are quite a few more). Adding old, obsolete models to such a study would be like adding a flat-earth model to a GCM ensemble – it makes little sense to consider models that have been shown to be inadequate. The situation with chill models is not the same as with GCMs – we do have a clear idea of which models are better, and there is really no rationale in my view to go for an ensemble approach.

*We admit that the methodology has probably not been adequately transmitted, as this is a key point of this work. The ensemble approach was only considered for climate models but results from the different chill models were considered, calculated and interpreted individually. The difference is that while chilling projections calculated with different climate projections and the same chilling model have been averaged, chilling projections from different chilling models were not. We have tried to clarify this point to avoid any impression of comparison between chilling model results in the results and discussion sections.*

*We are willing to discuss the validity of the models used. We agree with the first reviewer that there are several studies concluding that the Dynamic Model (DM) exhibits a higher accuracy than the Richardson based models (RbM, as Utah, North Carolina and De Melo-Abreu models). However, the reported improvement in the papers quoted by the referee is very small (e.g. Ruiz et al., 2007). Those studies also report varieties and locations where RbM models perform better. Also, some of these papers and others claim there is not a significant difference between models, for instance:*

*"In this study, **differences [between DM and Utah model] were not found** between these two models when estimating the chilling requirements for seven sweet cherry cultivars in north-western Murcia", Alburquerque et al. (2008), for cherry trees in Spain*

*"We have obtained **very homogeneous results with the Utah and Dynamic models**[...] The chilling requirements of the evaluated cultivars in the 3 years studied were quite homogeneous, according to the Utah and Dynamic models. Besides, the relationship between the two models was very close (R= 0.99).", Ruiz et al. (2007), for apricots in Spain. This team is also using Utah model for prune tree in Spain (Ruiz et al., 2015).*

*These results take part of the Luedeling et al. (2012) review; see Table 2, where for two studies in Spain DM appears with better performance, and for other two cases Utah model and DM appear with equivalent performance.*

*Therefore, in our view, although under for some cases DM has shown a better performance, we cannot conclude that DM is a better model for Spain in general terms.*

*This is also supported by other researchers using other models besides the DM in recent papers, as for instance (Darbyshare et al, 2013, made a study to evaluate the global warming on winter chilling in Australia using 0-7.2ºC, Modified Utah, and Dynamic Model; Miranda et al., 2013 compares Weinberger, Utah, North Carolina, Low Chilling and Dynamic for peach; Aybar et al., 2015, using a de Melo-Abreu model for analysing the suitability olives varieties in Argentina; Marra et al., 2017, using Richardson model and Chilling hours model, but no DM, for cherry in Italy; Sawamura et al., 2017, investigated the chilling requirements of peach in Japan using the Weinberger and Utah model). Also, the North Carolina model is currently being used by the Northeast Regional Climate Center from USA administration, implemented by the University of Cornell for apple tree (see below).*

[Figure]

*Source: http://www.nrcc.cornell.edu/industry/apple/apple.html*

*Therefore, we respectfully disagree with the referee's assessment of North Carolina model being obsolete as it is being currently used. Even the Chilling Hours Model, which is the oldest method to estimate winter chill accumulation (not considered in our paper), and considers all hours with temperatures ranging from 0 to 7.2 ºC equally effective, continues to be useful, as is still widely used in climate change impact and adaptation studies (see for example grapevine studies as Londo et al., 2014; Houston et al., 2018).*

*Additionally, if it was the case that DM is superior for our particular case, it would be important to notice that even with climate models, one could argue that some are non*

*adequate to reproduce specific (and also very important) climate aspects (e.g., the monsoon), but they are used anyway. We agree with the reviewer where he says (Luedeling et al, 2012): "All the models still leave a lot to be desired in terms of accuracy and some dormancy breaking behaviour at warm sites could not be explain at all".*

*Finally, we think that the main point is here is that all these models were developed, more than for specific locations, for specific tree species (peach for RbM). And the current practice is two or three models of chilling accumulation being used against phenological data of a specific species, generally with several varieties, and obviating that the model was fitted for a different crop (peach), assuming that there are not differences among species. In the few works where chilling accumulation models have been fitted for a different species than peach, differences respect to the fit for peach appeared. In our work, we have prioritized the adjustment parameters made to the RbM for different species (apple, which became North Carolina model; and olive, which became De Melo-Abreu model), under the hypothesis that the model would perform better if fitted to the behaviour of that species than if the model was is used with the parameters established for peach. In the case of the peach tree, the initial parameters of MbR and DM have been used.*

*Alburquerque, N., García-Montiel, F., Carrillo, A., and Burgos, L.: Chilling and heat requirements of sweet cherry cultivars and the relationship between altitude and the probability of satisfying the chill requirements, Environ. Exp. Bot., 64, 162-170, https://doi.org/10.1016/j.envexpbot.2008.01.003, 2008.*

*Aybar, V. E., Melo-Abreu, J. P., Searles, P. S., Matias, A. C., Del Rio, C., Caballero, J. M., and Rousseaux, M. C.: Evaluation of olive flowering at low latitude sites in Argentina using a chilling requirement model, Span. J. Agric. Res., 13, 10, 10.5424/sjar/2015131-6375, 2015.*

*Darbyshire, R., Webb, L., Goodwin, I., and Barlow, E. W. R.: Impact of future warming on winter chilling in Australia, International Journal of Biometeorology, 57, 355-366, 10.1007/s00484-012-0558-2, 2013.*

*Houston, L., Capalbo, S., Seavert, C., Dalton, M., Bryla, D., and Sagili, R.: Specialty fruit production in the Pacific Northwest: adaptation strategies for a changing climate, Clim. Change, 146, 159-171, 10.1007/s10584-017-1951-y, 2018.*

*Londo, J. P., and Johnson, L. M.: Variation in the chilling requirement and budburst rate of wild Vitis species, Environ. Exp. Bot., 106, 138-147, https://doi.org/10.1016/j.envexpbot.2013.12.012, 2014.*

*Luedeling, E.: Climate change impacts on winter chill for temperate fruit and nut production: A review, Scientia Horticulturae, 144, 218-229, https://doi.org/10.1016/j.scienta.2012.07.011, 2012.*

*Marra, F., Bassi, G., Gaeta, L., Giovannini, D., Palasciano, M., Sirri, S., and Caruso, T.: Use of phenoclimatic models to estimate the chill and heat requirements of four sweet cherry cultivars in Italy, Acta Hortic., 1162, 57-64, 10.17660/ActaHortic.2017.1162.10, 2017.*

*Miranda, C., Santesteban, L. G., and Royo, J. B.: Evaluation and fitting of models for determining peach phenological stages at a regional scale, Agric. For. Meteorol., 178-179, 129-139, https://doi.org/10.1016/j.agrformet.2013.04.016, 2013.*

*Ruiz, D., Campoy, J. A., and Egea, J.: Chilling and heat requirements of apricot cultivars for flowering, Environ. Exp. Bot., 61, 254-263, https://doi.org/10.1016/j.envexpbot.2007.06.008, 2007.*

*Ruiz, D., Egea, J., Salazar, J. A., and Campoy, J. A.: Necesidades de frío para la salida del letargo y necesidades de calor para florecer en variedades de ciruelo japonés (Prunus salicinia L.), XIV Congreso Nacional de Ciencias Hortícolas. SECH 2015. Retos de la Nueva Agricultura Mediterránea, Orihuela, Spain, 2015.*

*Sawamura, Y., Suesada, Y., Sugiura, T., and Yaegaki, H.: Chilling Requirements and Blooming Dates of Leading Peach Cultivars and a Promising Early Maturing Peach Selection, Momo Tsukuba 127, The Horticulture Journal, 86, 426-436, 10.2503/hortj.OKD-052, 2017.*

**3)** Related to the previous points, we've done several studies to compare the response of various chill metrics to climate change. First, they differ greatly in their sensitivity to warming (Luedeling et al., 2009c). Second, they are not comparable, with the ratio between different chill metrics varying tremendously across the globe, especially along climate gradients (Luedeling and Brown, 2011). Especially at the warmest end of the climatic range for temperate fruit trees, most models fail (Balandier et al., 1993; Benmoussa et al., 2017a, 2017b; Linsley-Noakes and Allan, 1994). The Dynamic Model is the only model I know that has a chance of somewhat describing changes correctly across different climates. This is the reason why in our 2011 paper (Luedeling et al., 2011) we only report Chill Portions (we actually calculated other metrics too, if I remember correctly, but I consider the results meaningless). This reasoning is actually described in several places in this paper and elsewhere (e.g. Luedeling, 2012). Just as an illustration, in the literature we found the chilling requirement of 'Ohadi' pistachios quantified at 1000+ CH in Turkey, but they grow well at 100 CH in Tunisia. This difference is not trivial at all and illustrates how badly off we can be if we use the wrong model.

- *With regard the comparison of various chill metrics:*

*We have introduced the reference Luedeling et al. (2009) as previous work on the comparison of response of various chill metrics to climate change. At the same time, we have stressed that this is not the objective of this paper and review the manuscript removing explicit and implicit comparisons between models. In fact, we have not*

*averaged results from different chilling models, keeping results separately, as explained in Answer#2. We have stressed it more in the paper, specifically in the Material and Methods section.*

- *With regard the performance of different models:*

*For a general answer, please see Answer#2. With regard models' performance in warm regions particularly, a worst performance is found not only for RbMs, but also for DM (Benmoussa et al., 2017 for pistachio in warm Sfax region in Tunisia):*
*https://www.sciencedirect.com/science/article/pii/S0098847217301119*
***"highlight: The Dynamic Model does not work well under Tunisian climate conditions."*** *This supports our argument that the performance of these models is not so different. Due to the lack of knowledge and data (especially for chilling portions) for accurate model calibration, including warm regions, we believe that uncertainty is better handled if not just one model is considered, even if they are not directly comparable.*

*Benmoussa, H., Luedeling, E., Ghrab, M., Ben Yahmed, J., and Ben Mimoun, M.: Performance of pistachio (Pistacia vera L.) in warming Mediterranean orchards, Environ. Exp. Bot., 140, 76-85, https://doi.org/10.1016/j.envexpbot.2017.05.007, 2017.*

*Luedeling, E., Zhang, M., Luedeling, V., and Girvetz, E. H.: Sensitivity of winter chill models for fruit and nut trees to climatic changes expected in California's Central Valley, Agric. Ecosyst. Environ., 133, 23-31, 10.1016/j.agee.2009.04.016, 2009.*

**4)** One particular criticism of chill models has been that they are calibrated for a particular site and not necessarily generally valid. There is a reason why the North Carolina Model and the Utah Model are named after geographic areas, not after crops, and why researchers in various places saw the need to make adjustments. For example, in South Africa the Utah Model regularly produced negative chill totals at the end of the season. This was 'addressed' by removing the chill negation (resulting in the Positive Utah Model: Linsley-Noakes and Allan, 1994). The necessity of these 'empirical hacks' clearly indicates that these models can't be trusted across climatic gradients – which is critically important for a credible climate change assessment.

*We agree with the referees 1 and 2 that, ideally, a site specific calibration would be desirable for any simulation exercise, as is the general practice in agronomic studies. As the second reviewer points out, indeed, the conditions of a calibrated model at one site do not completely coincide with those found in other locations. However, the state of the art of chill modelling is not yet there, and current practice is to apply these models elsewhere (see for instance many previous studies using these models in locations other than Utah, all of them without site-specific calibration, e.g. Alburquerque et al., (2008) for cherries in Spain; Razavi et al., (2011) for peach and Apricot in Iran; Sawamura et al., (2017) for peach in Japan). We think that in our case this is justified because in the model all the parameters that the researchers believe have relevance in the process are included. In our case, the main driver is temperature regime; and*

*actually, in the case of North Carolina model for apples, the main production area is Northern Spain, with climatological characteristics (temperature) more similar to North Carolina than the Spanish average. Accordingly, we have delimited more the concrete area of the apple tree production in the introduction section.*

*However, as we have discussed previously in Answer#2, we think that the main point here is that all these models were developed, more than for specific locations, for specific tree species (peach for RbM). And the current practice is two or three models of chilling accumulation being used against phenological data of a specific species, generally with several varieties, and obviating that the model was fitted for a different crop (peach), assuming that there are not differences among species. In the few works where chilling accumulation models have been fitted for a different species than peach, differences respect to the fit for peach appeared. In our work, we have prioritized the adjustment parameters made to the RbM for different species (apple, which became North Carolina model; and olive, which became De Melo-Abreu model), under the hypothesis that the model would perform better if fitted to the behaviour of that species than if the model was used with the parameters established for peach. In the case of the peach tree, the initial parameters of MbR and DM have been used.*

*This is a research gap indeed. As stated in Luedeling et al. (2011), estimates in Chill Portions (for the Dynamic model) are less widely available than estimates in other metrics, and although if the knowledge gap in that sense have been reduced nowadays, estimates for many crops and varieties are still not available. We agree that, ideally, more experimental data should be generated to improve the chilling simulation, not only because of the differences between locations, but mainly due to the huge uncertainty related to the species and variety requirements, that in our view, is much more important than that related to the models. We agree that is a scientifically relevant issue, so we have included a comment on this on the discussion to raise awareness on the referee's point.*

*Alburquerque, N., García-Montiel, F., Carrillo, A., and Burgos, L.: Chilling and heat requirements of sweet cherry cultivars and the relationship between altitude and the probability of satisfying the chill requirements, Environ. Exp. Bot., 64, 162-170, https://doi.org/10.1016/j.envexpbot.2008.01.003, 2008.*

*Luedeling, E., Girvetz, E. H., Semenov, M. A., and Brown, P. H.: Climate Change Affects Winter Chill for Temperate Fruit and Nut Trees, PLOS ONE, 6, e20155, 10.1371/journal.pone.0020155, 2011.*

*Razavi, F., Hajilou, J., Tabatabaei, S., and Dadpour, M.: Comparison of Chilling and Heat Requirement in Some Peach and Apricot Cultivars, Research in Plant Biology, 1, 40-47, -, 2011.*

*Sawamura, Y., Suesada, Y., Sugiura, T., and Yaegaki, H.: Chilling Requirements and Blooming Dates of Leading Peach Cultivars and a Promising Early*

*Maturing Peach Selection, Momo Tsukuba 127, The Horticulture Journal, 86, 426-436, 10.2503/hortj.OKD-052, 2017.*

**5)** The presumably innovative outlook of possibly using estimates of the amount of chill that is exceeded 90% of the time (p. 10, l. 29) isn't so innovative after all. In fact, we already used this 'Safe Winter Chill' approach in several publications, dating back to 2009 (Luedeling et al., 2009a, 2011). It has also been picked up by others (though I don't currently remember who that was).

*The novelty was referred to the EOA index application (see Rodríguez et al., 2019) to analyse the robustness of projections of having a safe winter chill. In other words, it refers to the robustness metric (the EOA index) application, not to the safe winter chill definition, which is used only as the hypothesis for the EOA index. We have reformulated the sentence in the further work paragraph to make it clearer. Also, we have added a quotation (Luedeling et al., 2009) wherever in the manuscript that reference to safe winter is done.*

*Luedeling, E., Zhang, M., and Girvetz, E. H.: Climatic Changes Lead to Declining Winter Chill for Fruit and Nut Trees in California during 1950–2099, PLOS ONE, 4, e6166, 10.1371/journal.pone.0006166, 2009.*

*Rodríguez, A., Ruiz-Ramos, M., Palosuo, T., Carter, T. R., Fronzek, S., Lorite, I. J., Ferrise, R., Pirttioja, N., Bindi, M., Baranowski, P., Buis, S., Cammarano, D., Chen, Y., Dumont, B., Ewert, F., Gaiser, T., Hlavinka, P., Hoffmann, H., Höhn, J. G., Jurecka, F., Kersebaum, K. C., Krzyszczak, J., Lana, M., Mechiche-Alami, A., Minet, J., Montesino, M., Nendel, C., Porter, J. R., Ruget, F., Semenov, M. A., Steinmetz, Z., Stratonovitch, P., Supit, I., Tao, F., Trnka, M., de Wit, A., and Rötter, R. P.: Implications of crop model ensemble size and composition for estimates of adaptation effects and agreement of recommendations, Agric. For. Meteorol., 264, 351-362, https://doi.org/10.1016/j.agrformet.2018.09.018, 2019.*

**6)** Another alleged innovation is the variable duration of the chilling period, which is determined by the minimum and maximum chill accumulation. Sure, this is new, but is it correct? The authors don't present any evidence for this. I realize that some authors have claimed that something like this makes sense (e.g. Cesaraccio et al., 2004 for their own model, but others have also said this for the Utah Model I think), but is there really any evidence? Actually, I strongly doubt that trees can make use of chill accumulation over the entire cold period. We've done a number of studies where we tried to statistically determine the chill-responsive period (Guo et al., 2015; Luedeling and Gassner, 2012; Luedeling et al., 2013a, 2013b), and we've always found periods that are much shorter than the full winter season. Now this may mean various things, including that trees are pretty safe from chill shortfalls in many places, but I suspect that it would make sense to end the chilling period earlier than an automatic algorithm

would suggest (actually, if I could change one thing about our earlier studies, I would shorten the period we considered, which seems much too long now in hindsight).

*The referee's discussion and the studies quoted (Guo et al., 2015; Luedeling and Gassner, 2012; Luedeling et al., 2013a, 2013b), in our view, reflect that there is a lot to learn about how trees work in relation to chilling accumulation. We agree that it is reasonable to question if trees can make use of the whole chilling accumulation period, and we have commented this fact about the possibility of an overestimation of the chilling accumulation in the discussion.*

*At the same time, we have decided not to choose a fixed period approach. On the one hand, a fixed starting date and duration for the chilling period for sure will introduce errors, as every year is different for every location and for every climate model. Some studies use self-regulating dates (we have quoted them)for chill models because of the lack of reliable physiological markers and the inefficacy of fixed dates to account for the mentioned seasonal climate variability (Measham et al. 2017). For instance, Marra et al. (2017), where an approach to calculate the starting date, using a self-regulating algorithm similar than in the present study, found that the applied method allowed a significant improvement compared to other studies that fix the date at October 1st. Also, results in the Measham et al. (2017) study show a larger variability in the chilling portions accumulation using a fix dates approach than a self-regulating one, as some chilling portions were excluded due to a late initial date. On the other hand, we have decided not to select a fix final date, even when it could be very well defined, because it will become eventually meaningless in a climate change context. A fixed period would cause a lot of problems and inconsistencies when the cold period is clearly shifted along the year at the end of the century.*

*Other argument that supports the use of a self-regulating method is that changes in chilling projections become very much comparable among different methods and with the present, even when having in mind the possible overestimation mentioned by the referee.*

*Marra, F., Bassi, G., Gaeta, L., Giovannini, D., Palasciano, M., Sirri, S., and Caruso, T.: Use of phenoclimatic models to estimate the chill and heat requirements of four sweet cherry cultivars in Italy, Acta Hortic., 1162, 57-64, 10.17660/ActaHortic.2017.1162.10, 2017.*

*Measham, P. F., Darbyshire, R., Turpin, S. R., and Murphy-White, S.: Complexity in chill calculations: A case study in cherries, Scientia Horticulturae, 216, 134-140, https://doi.org/10.1016/j.scienta.2017.01.006, 2017.*

**7)** The paper starts with a strange introduction about the classification of fruit trees, which I'm not sure I agree with and which is also not relevant here. This paper is only about temperate species, so no need for such a general take. The first two paragraphs should be deleted.

*Our attempt was to take into account that this journal serves to a wide and diverse community of readers (as stated in the NHESS journal aims and scope) with this general introduction. However, we have reduced and focused it following referee's suggestion. First two paragraphs have been removed.*

**8)** I strongly urge the authors to make their code public, either in a repository or as supplementary materials to this paper. This will make it much easier to understand what was done. For instance, the statement that the authors used the method by Fishman et al. (1987a, 1987b) is not sufficiently detailed – anyone who's seen these papers knows that this is not at all trivial to implement (and I wonder if this is really the authors' source of the algorithm). Ideally, a paper should be reproducible, meaning that the methods should be sufficiently detailed for readers to repeat an experiment. This is often not really achievable, but it is not difficult for a modeling study such as the one described here. Please share the code. The main reason for this is that the actual results of this paper are not particularly helpful – pretty much the same has been shown before. The innovation (for the chill modeling community) lies in the climate data processing, but if this isn't actually shared with readers, nobody can easily make use of this methodology. In my view, the offer that readers can contact the authors isn't sufficient.

*All the algorithms used in this paper have been programmed, implemented and executed by the authors. In our team we have experts from different fields, being a computer engineer one of them. The implementation of the model was done by using the model constants commonly used in standard applications, following other studies like Luedeling et al., (2011). We have included the reference in that sense as it has been followed the same procedure.*

*We chose to share the code by the formula "under request and quotation", that means a simple e-mail message of request without further registration, as our institution recommends to do so, to keep track of the research groups and publications derived using it. This is the case of many software developments (e.g. DSSAT source code available upon request).*

*However, as both referees raised this point, we have included the code as supplementary material. Specifically, we have included: chilling model codes, hourly temperature calculation and chilling computation period for the RbM models.*

*Luedeling, E., and Brown, P.: A global analysis of the comparability of winter chill models for fruit and nut trees, 411-421 pp., 2011.*

**9)** Finally, I suggest that the authors compare their results (and maybe also their methods) with similar studies that have been done before. There have been quite a

few, as the authors will realize if they do a systematic search, not necessarily on Spain, but on various other regions.

*We have compared our results with the references included in the Answers#1, 2, 3, 4 and 6, and with Luedeling et al. (2009a and 2009b) for California and Darbyshire et al. (2013) for Australia, which are particularly interesting for us because they were conducted in regions with Mediterranean climate. This has been done in the discussion section.*

*Darbyshire, R., Webb, L., Goodwin, I., and Barlow, E. W. R.: Impact of future warming on winter chilling in Australia, International Journal of Biometeorology, 57, 355-366, 10.1007/s00484-012-0558-2, 2013.*

*Luedeling, E., Zhang, M., and Girvetz, E. H.: Climatic Changes Lead to Declining Winter Chill for Fruit and Nut Trees in California during 1950–2099, PLOS ONE, 4, e6166, 10.1371/journal.pone.0006166, 2009a.*

*Luedeling, E., Zhang, M., Luedeling, V., and Girvetz, E. H.: Sensitivity of winter chill models for fruit and nut trees to climatic changes expected in California's Central Valley, Agric. Ecosyst. Environ., 133, 23-31, 10.1016/j.agee.2009.04.016, 2009b.*

**10)** Even more finally, I suggest language editing. There is still some room for improvement in terms of language, and some statements are unclear.

*The manuscript was edited by a professional language service previous to submission (the invoice will be privately sent to the editor due to data protection). The same service will be used on the revised manuscript if required.*

**Minor issues:**

p. 1, l. 14: what are 'inner physiological factors'?

*Lang et al., (1987) defined endodormancy as that which is regulated by physiological factors inside the affected structure. It is a definition widely used. We have included the definition instead the expression 'inner physiological factors'.*

*Lang, G. A., Early, J. D., Martin, G. C., and Darnell, R. L.: Endo-, para-, and ecodormancy: physiological terminology and classification for dormancy research, HortScience, 22, 371-377, 1987.*

p. 1, l. 14: 'accumulating cool temperatures to finish dormancy is unclear (at least in terms of what dormancy this is – I'd most likely associate finishing dormancy with bloom of leaf out, but that also requires heat). No need for "be broken" in quotation

marks. This is commonly used and doesn't need to be identified as an odd term (or whatever the purpose of the quotation marks is).

*Quotation marks have been removed, and the sentence have been reformulated as follows:*
*"accumulating chilling temperatures to finish this sort of dormancy".*

p. 1, l. 16: I don't think the chilling requirement is different for each variety (which means that no two varieties have the same requirement). They are crop and variety-specific, but not all different.

*Yes, the referee is right. We have modified the sentence to avoid this possible misunderstanding, as follows:*
*"chilling accumulation required to break dormancy depends on specie and variety"*

p. 1, l. 28 – p.2, l. 10: irrelevant – delete
*We have deleted the sentence.*

p. 2, l. 12: income, not wealth
*Yes, the referee is right. We have modified the sentence as suggested.*

p. 2, several places: for simplicity and reader-friendliness, I recommend replacing 10ˆ6 by 'million'
*We have modified the sentence as suggested.*

p. 2, ll. 18-19: FAOSTAT doesn't directly provide such values I believe, so it would be important to state how this was determined (also note that there are all kinds of issues with this database). It is also not obvious that this sentence refers to the global scale, since the previous sentence talks about Spain. Overall, this isn't a very relevant statement in a paper that's just on Spain.

*In the FAOSTAT /Data/Crops webpage, it is possible to select a crop and gather worldwide data for a particular crop. According to those data, Spain is a major fruit producer in the world and, consequently, studies on Spain are relevant. We have briefly mentioned the process we followed to obtain the showed information from FAOSTAT service in the text.*

p. 2, l. 24: I believe the thing trees are sensitive to is frost (not generally cold temperatures)

*Yes, the referee is right. We have modified the sentence as suggested.*

p. 3, l. 1: 'accumulation of cold periods' is an unfortunate choice of words. Sounds like trees need, say, 5 cold periods to break endodormancy.

*Yes, the referee is right. We have reformulated it for making it clear as follows:
"the accumulation of time exposed to cold temperatures"*

p. 3, l. 3: not all models are based on temperatures between certain thresholds. The Dynamic Model works differently, and even the Utah-type models don't really follow this simple structure.

*Yes, the referee is right. We have modified the sentence by removing
"all based on the accumulated time with temperatures between certain thresholds".*

p. 3, 12: I disagree that the chilling requirement corresponds to conditions where a tree is grown. It may rather correspond to conditions where it evolved/was bred

*Yes, the referee is right. We have modified the sentence as follows:*

*"Each tree species and variety has specific chilling requirements for correct plant development, usually related to the environmental conditions where it evolved or was bred".*

p. 3, ll. 13-17: not sure what information is conveyed here. The initial statement is about considering a range, but then the examples are precise values, not ranges. If this is supposed to illustrate intra-specific variation, then please make sure to use the appropriate terminology (not sure what 'crop tree' refers to).

*Yes, we understand the referee's point. We have modified the text as follows:
"As a result, for a given species a range of estimates of chill accumulation encompassing all varieties has to be considered. For instance, for the apricot varieties considered in Campoy et al. (2012), the estimated accumulated chilling varies between 413 ('Palsteyn' variety) and 1172 ('Orange red' variety) chill hours (chilling hours method). This range is 613-777 when chilling units by Utah method are computed, and 37-64 chill portions when Dynamic method is applied."
Also, we have replaced the expression "crop tree" by "fruit tree" throughout the paper.*

p. 4, l. 9 (or elsewhere): Somewhere the authors need to mention the various chill assessments that have already been done by a number of people in a wide range of places.

*Yes, the referee is right. We have mentioned the references listed in the answers to major issues (above in this document) in several parts of the text.*

p. 4, l. 17: no, the models do not need hourly Tmin and Tmax. They just need hourly temperature, which can be derived (if no other information available) from daily Tmin and Tmax.

*Yes, the referee is right. We have modified the sentence as follows:*

*"The climate variable required by the chilling models used in this study is hourly temperature, which can be derived, when no other information is available, from minimum (Tmin) and maximum (Tmax) temperatures."*

p. 4, l. 22: not sure what 'freely distributed' means. Open-access?

*We have used the exact term used by the creators of the dataset (http://www.meteo.unican.es/datasets/spain02)*
*It means that you can download the data with the condition of quoting two references provided. We have clarified it in the text specifying that free downloading is possible.*

p. 4, l. 24: is this really an observational dataset?

*Yes, it is. The methodology for generating these databases is robust and widely known on climate modelling studies: direct observations are interpolated in a physically-based way to a regular grid to be usable for climate models' comparison purposes. For instance, E-OBS (Haylock et al., 2008) and CRU (Harris et al., 2014) databases were built using this methodology.*

*Also, please see the link in the previous comment, where the database is described. Also, the quote Herrera et al., 2016 title reads:*
*Herrera et. al. (2016) Update of the Spain02 Gridded **Observational** Dataset for Euro-CORDEX evaluation: Assessing the Effect of the Interpolation Methodology. International Journal of Climatology, 36:900–908. DOI: 10.1002/joc.4391.*

*We have added the link (http://www.meteo.unican.es/datasets/spain02) in the text.*

*Harris, I., Jones, P. D., Osborn, T. J., and Lister, D. H.: Updated high-resolution grids of monthly climatic observations – the CRU TS3.10 Dataset, Int. J. Climatol., 34, 623-642, 10.1002/joc.3711, 2014.*

*Haylock, M. R., Hofstra, N., Klein Tank, A. M. G., Klok, E. J., Jones, P., and New, M.: A European daily high-resolution gridded dataset of surface temperature and precipitation, D20119 pp., 2008.*

*Herrera et. al. (2016) Update of the Spain02 Gridded Observational Dataset for Euro-CORDEX evaluation: Assessing the Effect of the Interpolation Methodology. International Journal of Climatology, 36:900–908. DOI: 10.1002/joc.4391.*

p. 5, l. 15: more details are needed on the temperature generation, especially since the source will be hard to find for most readers. What mathematical functions were used for constructing daily curves? The common method in horticultural studies such as this

one is a methodology by Linvill (1990), which is based on a sine curve during the day and logarithmic cooling at night (implemented in the chillR package; Luedeling, 2018). I'd be quite curious to learn how de Wit's method compares with this, but the authors provide insufficient information about their approach.

*Yes, we used de Wit's method. MATLAB code has been made available in the supplementary material as requested by the referee.*

p. 6, ll. 11-13: The authors compute a mean and then a median. Later in the paper they argue that one should calculate a 10% quantile. Why didn't they do this here?

*The objective of this paper as stated in page 4 line 10 is to assess the impact of climate change on temperate fruit tree chilling accumulation Spain. This general objective is better achieved by an averaged indicator, as median and mean. The suggestion of using the 10th quantile was only introduced in page 10, starting from line 20, proposals for further work, consisting in using the EOA index for analysing chances of robust, high confidence, local adaptation. This EOA index needs a threshold definition, for which we propose the 10th quantile (so we do not need nor suggest using it for other purpose than that). This is a refinement of previous assessment of average impact, but we consider this further work out of the scope of the current study.*

*We have modified the text to make this point clearer.*

p. 6, ll. 16-17: As stated above, I'd prefer to have the code made publically available, for full transparency and usefulness.

*Please see the Answer#8. Codes have been provided as supplementary material.*

p. 6, l. 23: Is the full name of MAPE really 'mean percentage absolute error'? That would seem to lead to the acronym 'MPAE'

*Yes, you are right, this is a typo. That line has been changed by "mean absolute percentage error". In other parts of the document (i.e. page 19, line 6) the order is correct.*

p. 7, l. 19: 'similarly simulated' is awkward wording

*It has been changed to "simulated in a similar way".*

p. 7, ll. 23-27: All these models use different units, so they can't be compared (the fact that they're probably all called chill units doesn't make them equivalent). While it's obvious that the Dynamic Model can't be compared to the others (because values are much smaller than for the other metrics), the others are also not comparable!

*Yes, we understand that the reader could interpret that the models with the same units could be comparable. We have modified the text to clarify these aspects.*

p. 8, l. 27: scenarios were averaged in this study, but we also provided information for determining the impact of climate model and emissions scenario.

*We have modified this sentence as follows:*

*"The chilling portion results are in agreement with the projections from Luedeling et al. (2011) in the Mediterranean region for different periods, where emission scenarios and global climate models were averaged (see Fig. 6 in Luedeling et al., 2011; information for determining the impact of climate model and emissions scenario was provided in that study)."*

p. 9, ll. 1-2: As stated above, I don't consider it an asset to include outdated models in a study…

*Please see Answer#2 to major issues above in this document.*

p. 9, l. 22: not sure what 'discrete nature' means. And I also think that this may be an indication that these models are too sensitive for warm places.

*We meant discrete as opposite as continuous. We think that the high values of CV are related to the low values of the chilling in absolute terms, which actually is in agreement with referee's suggestion: this might makes these models too sensitive for warm places. We have included this explanation in the discussion.*

p. 9, ll. 26: this study didn't 'find' this, it just reported on it. Luedeling et al. (2011) sort of found this.

*We have modified this as suggested, using the verb "report".*

p. 10, ll. 4: Yes, it would be great to have more datasets, but we actually already have a lot. Rather than call for collecting more data, I'd call for better use of such data for model development and validation.

*Probably the referee is right and it is more about data availability and access and less about data existence. At least in the case of Spain, although it is true that there are several works on the subject, there are species/varieties with little data availability and the models developed up to now have important shortcomings. We have specified that the scarcity mentioned in the paper is referred to the available data in Spain, as we rely on referee's knowledge about elsewhere.*

p. 10, ll. 11-12: 'crop varieties depending on the RCP' is unfortunate wording. First, crop varieties don't depend on RCP. Second, RCPs are theoretical pathways that not be followed precisely. Better to say something like 'depending on how rapidly GHG emissions can be reduced' or something like that.

*Yes, we understand how the sentence could be misunderstood. We have modified it according referee's suggestion.*

p. 10, l. 23: not sure what 'low-limit chill requirements' are

*We meant the variety with the lower chilling requirement within a given species. We have used that expression to make the sentence more understandable.*

p. 10, l. 29: as mentioned above, this is exactly what the Safe Winter Chill metric achieves.

*Yes, we are referring to that, we have introduced a quote here (Luedeling et al., 2009)*

p. 11, 2-4: It's obvious that RCP8.5 causes greater change, similar to the end vs. middle of century. Doesn't need to be mentioned or should clearly be marked as expected.

*Text has been modified as suggested.*

p. 11, l. 6: why especially in warm regions? The impact depends not only on chill loss, but also on what is grown there and how much chill it needs.

*The text has been modified as follows:*

*"A winter chill reduction may threaten the viability of some crops and varieties, especially in some areas that already have a low number of chilling units and are cultivated with chilling demanding species, where their reduction may jeopardise the cultivation of some tree crops within the near future."*

p. 11, ll. 17-18: confusing sentence.

*The text has been modified as follows:*

*"Such an adaptation would benefit from mitigation, as adaptation is assumed to be more feasible for moderate warming scenarios."*

Reference list: It would be so much easier to look through this, if all but the first row of a reference were indented.

*The section we has been modified as suggested.*

Maps: maps should have a coordinate system, north arrow, scale bar etc.

*We have included the suggested information in the corresponding figures.*

Fig. 1: I doubt that all the olive data are right. If so, some parts of Spain would be almost exclusively olives.

*We have checked the data and they are correct. Source is the Spanish Ministry related to agriculture and official statistics. Jaen province (Andalusia) is the largest area of olive trees in the world. When travelling through it (simply from the highway) you can only see olive trees for kilometres (please see image below).*

[Figure]

Source:https://www.google.es/maps/@37.6076977,-4.0473674,3a,60y,283h,73.24t/data=!3m6!1e1!3m4!1sTVFJSEzMRW_Jco1F645SpA!2e0!7i13312!8i6656)

Maps 3-7: very hard to compare changes, which is really the most important part of this paper, if the maps are scattered across various places.

*We have rearranged the figures to bring map of change together.*

Fig. 5: is the scale used for the change useful.

*We have adapted the scale to the new figures, and we have tried to make it useful.*

In summary, I think this contribution has potential, since the way the climate data were processed is very robust. But the team should consider adding some chill modelling capacity to the study to make this more convincing. While chill seems like an easy application of a climate change projection framework (it's assumed to just depend on temperature after all), things are actually quite complicated due to the invisibility of chill induced changes, which has precluded the development of convincing models so far. In consequence, there are many models, and most of them are not suitable for studies across climates. If the authors manage to adequately consider this, this manuscript may become publishable.

*Thank you for your thoughtful revision. We have addressed the issues summarized by the referee in the answers above. We are convinced that our arguments are correct and sound, but if the editor and both referees ask us to remove some of the chilling models considered, we would be willing to do so.*

**Materials and methods**

Regarding the selection of models and scenarios, although hardly done in literature, the choice of models could be better justified using methodologies as in (Mendlik and Gobiet, 2016), since there is evidence of high sensitivity of climate model selection (Wilcke and Bärring, 2016). However, the authors chose the two reasonable scenarios (RCP 4.5 and RCP 8.5), allowing for consistent comments on importance of mitigation in context of actual discussion. Key equations of the chilling models should be provided in the additional material. In the main text a comment on the validation of the models should be given, in the view of their applicability on future time series.

*The ensemble of climate models contains 10 members, which was the whole set of models available. Additionally, Figure 3 is meant prove that our inputs are robust. To take into account the referee's concern we have included the following (bold) text in the corresponding section:*

*"..... This ensemble size is considered to be large enough by the agricultural impact community to retrieve robust results (Martre et al., 2015; Rodríguez et al., 2019).* **Due to the complex orography of the Iberian Peninsula and its remarkable climatic diversity (CLIVAR-Spain, 2010), no additional systematic selection was performed to reduce the number of RCM ensemble members (e.g., Mendlik and Gobiet, 2016). A thorough analysis in this sense would imply decomposing the Iberian Peninsula into several climatic sub-regions (Wilcke and Bärring, 2016) and would derive into a much more complex process to potentially improve**

*already robust results. The outputs of the EUR-11 ensemble for two RCPs were considered: 1) +4.5 W/m2 radiative forcing increase at the end of the 21st century relative to pre-industrial levels (RCP4.5) and 2) the same but for +8.5 W/m2 (RCP8.5)."*

*CLIVAR-Spain: Climate in Spain: past, present and future. Regional climate change assessment report, Ministerio de Ciencia e Innovación (España), Ministerio de Medio Ambiente y Medio Rural y Marino (España) 978-84-614-8115-6, www.clivar.es, 2010.*

*Mendlik, T., and Gobiet, A.: Selecting climate simulations for impact studies based on multivariate patterns of climate change, Clim. Change, 135, 381-393, 10.1007/s10584-015-1582-0, 2016.*

*Wilcke, R. A. I., and Bärring, L.: Selecting regional climate scenarios for impact modelling studies, Environ. Modell. Softw., 78, 191-201, https://doi.org/10.1016/j.envsoft.2016.01.002, 2016.*

**Results**

With regards to the CV, MAPE and IQR, the classes > 20, >0.4... are in my view not informative enough. Also, in section 3.1, the MAPE values are declared as problematic above 20% for few grid points, without mentioning until how high they stretch. Thus, no conclusion can be made if the computation for these grid points can be trusted at all.

*High MAPE values can be related also to the low values of the chilling accumulation in those areas, and therefore it does not mean that necessarily the projections cannot be trusted at all. It means that we should be more careful when interpreting the results. Nevertheless, we have marked somehow the areas in the plots where values were greater than 20%. Also, we have chosen a more understandable, representative classes for the figures, and the top end has been specified.*

**Discussion**

The difference between the two researched scenarios could be expressed more clearly (p.10, ll.11-19).

*We have introduced a sentence here discussing the main difference found between results obtained for each RCPs.*

- *On the models to be reported: Please see our answers above. We are convinced that our arguments are correct and sound, but if the editor and both referees ask us to remove some of the chilling models considered, we would be willing to do so.*

Figure 1 shows a good overview of land use in Spain for the reader, exposing major growing areas for the considered crops. Values seem reasonable from my experience. However, the choice of the color map is unfortunate, <1%, which could be conceptually be negligible, is very hard to distinguish from the higher classes. I suggest to revise the classification to a lower number of classes, 5 being preferred. A clarification is needed whether the map shows the percentage from the total area or from area classified as cropland.

*We have modified the figure 1 as suggested. We have specified that the percentage refers to the total area.*

Figure 2 features a useful example output of the analysis, but it was not justified that this is a representative example. The most reliable model would have been preferred, the Dynamic model was judged as best performing (Luedeling, 2012). In subplot B, over the years, the chilling units decrease, a trend line could be interesting, next to the mean. Subplot C should highlight which model is used for subplots A and B. In subplot D, neighboring grid points expose substantial differences in this mountainous terrain. With regards to the shortcomings mentioned in mountains areas, a further study could envision a more focused analysis on those areas.

*The example was considered representative for two reasons, 1) because it shows the general procedure followed for each cell to obtain an individual outcome from the climate ensembles, illustrating how the methodology aggregated yearly information and projections using different climate models, 2) to explain how the initial and final chilling accumulation dates are calculated, and this is particularly important for the Utah-based methods considered in the study as warm temperatures sometimes negatively contribute to chilling accumulation. This is not the case of the Dynamic model where only positive increases of chilling accumulations are added up, being the purpose of the cited figure of illustrating one of the Utah-based methods, more complicated to explain in that sense. So Dynamic model is not as useful as the others to illustrate this.*

*The text has been changed to further explain the need of this figure and the footnote has been modified to clarify that all subplots refer to the same model.*

*We agree with the referee's suggestion that a further study more focused on mountainous areas would be very much interesting from the scientific point of view. However, our priority was to focus in main productive areas that are usually at lower regions.*

Regarding Figures 3 -8 and as mentioned above, classes such as >20 are little informative. In this line, it would be of great value if the maps could either exclude or highlight less reliable outcomes. This could be done by keeping grid points white, or, if readability is not compromised, with a hatched overlay. From visual comparison, there seems to be a substantial part of the apple cultivation shown in Figure 1 in coastal and mountainous areas, those reported as with comparatively high errors.

*As answered above, we have chosen a more understandable, representative classes for the figures, and we have highlighted the areas >20 in these figures to facilitate interpretation.*

Technical comments (additionally to those mentioned in RC1, to which I fully agree):

* P.2 l. 26 delete 'it

*We have deleted it.*

* P.2 l.18, production, not productivity (if productivity is meant, the reference i.e. area should be specified, and I agree it is not relevant in this paper, rather give the importance of other fruits in Spain, ideally with national statistics rather than FAOSTAT)

*Yes, the referee is right, we have changed it in the revised version.*

* P.3 l.34, add 'among other regions'

*The referee's suggestion has been included in the revised version.*

\* P.5 l.23 inconsistent usage of Dynamic model / Dynamic method

*The referee's suggestion has been included in the revised version, using Dynamic model throughout the paper.*

\* P.8 l.10, specify where the biggest change occurred

*We have specified it in the revised version.*

\* P.9 l.16, Mediterranean','

*The referee's suggestion has been included in the revised version.*

\* P.9 ll.17-18 reformulate

*The referee's suggestion has been included in the revised version as follows:*

*"In light of the results, our hypothesis is that the stations in these areas are poorly represented by the interpolated Spain02 dataset."*

\* P.9.l.21 a warmer scenario

*The referee's suggestion has been included in the revised version.*

\* P.9 ll.28-29 'Nonetheless, few tree crops are grown [...]' – have these areas also be found as potential new cropping areas?

*Yes, at the lower part of the mountains, but the affected areas would be relatively small. That is why in our view improving the estimations for these areas would be interesting of course but not a priority.*

\* P.10 l.17 are you comparing this value (469 chilling units, according to the De Melo-Abreu method) with all outputs? It should only be compared to the output of the analysis using the same method, which, in the case of the far future under RCP8.5, where the map shows mainly values between 500-1000 chill units in the area coinciding with olive cropping.

*The comparison is established only between results from the same models. We have stressed this in the revised version as it is a key point. it seems that it was not clear*

*enough. Also, we have specified more the region we are referring to (red areas in Figure 8, first column, third row).*

*Thank you for your thoughtful revision. We have tried to address all the issues you raised. We are convinced that our arguments are correct and sound, but if the editor and both referees ask us to remove some of the chilling models considered, we would be willing to do so.*

**RCP4.5**

(a) EM   (b) CHANGE

**RCP8.5**

(c) EM   (d) CHANGE

Utah

North Carolina

De Melo-Abreu

Dynamic

**Chill units**
- <500
- 500-1000
- 1000-1500
- 1500-2000
- 2000-2500
- 2500-3000
- 3000-3500
- >3500

**Chill portions**
- <25
- 25-50
- 50-75
- 75-100
- 100-125
- 125-150
- 150-175
- >175

**Chill change**
- <-1200
- -1200--900
- -900-600
- -600--300
- -300-0
- >0

**Chill change (portions)**
- <-60
- -60--45
- -45--30
- -30--15
- -15-0

0  230  460    920
Kilometers

N

~~Figure 5. Maps of chilling accumulation, inter-annual variability and uncertainty results for RCP4.5 scenario in the NF (2021-2050). For each chilling method (rows), the median of the 30-year mean chilling sums of the 10 EUR-11 ensemble members (first column), change in chilling accumulation with respect to the baseline period (second column), the 10 EUR-11 ensemble members' mean coefficient of variation (CV, expressed per unit) of the 30-year period (third column) and the ensemble's interquartile range (IQR) of the 30-year mean chilling sums of the 10 EUR-11 ensemble members (fourth column).~~

**Figure 5. Maps of chilling accumulation in the 2021-2050 period. For each chilling model (rows), the median of the 30-year mean chilling sums of the 10 EUR-11 ensemble members and change in chilling accumulation with respect to the baseline period, for RCP4.5 scenario (first and second column respectively) and RCP8.5 scenario (third and fourth column respectively). Grid cells**

with mean absolute percentage error (MAPE, %) values from the validation phase higher than 20 are not considered as reliable results and are highlighted in diagonal lines.

[Figure]

RCP8.5 2021-2050

[Figure]

**Figure 6.** Same as Fig. 5 but for the FF (2071-2100 period).

**RCP4.5  2071-2100**

[Figure]

(a) EM    (b) CHANGE    (c) CV    (d) IQR

Utah   North Carolina   De Melo-Abreu   Dynamic

**Chill units**
- <500
- 500-1000
- 1000-1500
- 1500-2000
- 2000-2500
- 2500-3000
- 3000-3500
- >3500

**Chill portions**
- <25
- 25-50
- 50-75
- 75-100
- 100-125
- 125-150
- 150-175
- >175

**Chill change**
- <-1200
- -1200--900
- -900-600
- -600--300
- -300-0
- 0-300
- 300-600
- 600-900
- >900

**Chill change (portions)**
- <-60
- -60--45
- -45--30
- -30--15
- -15-0

**CV**
- 0-0.1
- 0.1-0.2
- 0.2-0.3
- 0.3-0.4
- >0.4

**IQR**
- 0-100
- 100-200
- 200-300
- 300-400
- >400

**IQR (portions)**
- 0-5
- 5-10
- 10-15
- 15-20
- >20

**RCP8.5 2071-2100**

[revised manuscript text omitted]

Figure 2S. Map of areas with chilling accumulation, in the 2071-2100 period and for RCP8.5 scenario, projected to be higher (green grid cells) or lower (red grid cells) than 1050 chilling units (chilling requirements estimated for Golden Delicious apple variety with the North Carolina model, plot a), than 469 chilling units (chilling requirements estimated for Picual olive variety with the De Melo-Abreu model, plot b), than 813 chilling units (chilling requirements estimated for Redhaven peach variety with the Utah model, plot c) or than 73 chilling portions (chilling requirements estimated for Redhaven peach variety with the Dynamic model, plot d). Grid cells with mean absolute percentage error (MAPE, %) values from the validation phase higher than 20 are not considered as reliable results and are highlighted in diagonal lines.

**Code 1S. Function *chill portions* MATLAB code, for calculating the chilling portions accumulated by the Dynamic model (Fishman et al., 1987a; Fishman et al., 1987b), with the standard parameters and the equations following Luedeling and Brown (2011)**

```matlab
function [delt] = chill_portions(t,time)

% Julian day to cut between seasons
DIA_CORTE = 186;

% See the "cuts" now in hourly approach
pos = [1];
for i=min(time(:,1)):max(time(:,1))
    x = time(:,1) == i;
    aux = find(x);
    aux = aux((DIA_CORTE-1)*24+1:end);
    if ~isempty(aux)
        pos = [pos aux(1)];
    end
end

% Constants
e0 = 4153.5;
e1 = 12888.8;
a0 = 139500;
a1 = 2567000000000000000;
slp = 1.6;
tetmlt = 277;
aa = a0/a1;
ee = e1-e0;

tk = t+273.15;
ftmprt = slp*tetmlt*(tk-tetmlt)./tk;
sr = exp(ftmprt);

xi = sr./(1+sr);
xs = aa*exp(ee./tk);

ak1 = a1*exp(-(e1./tk));
inters = zeros(size(ak1));
intere = zeros(size(ak1));
delt = zeros(size(ak1));
portions = zeros(size(ak1));

for i=1:length(inters)
    if ismember(i,pos)
        inters(i)=0;
        delt(i)=0;
```

```matlab
                portions(i) = 0;
            else
                if intere(i-1)<1
                    inters(i) = intere(i-1);
                else
                    inters(i) = intere(i-1)*(1-xi(i));
                end

            end
        intere(i)= xs(i)-(xs(i)-inters(i))*exp(-ak1(i));

        if ~ismember(i,pos)
            if intere(i)<1
                delt(i) = 0;
            else
                delt(i) = xi(i)*intere(i);
            end
            portions(i) = delt(i) + portions(i-1);
        end
    end

end
```

**Code 2S. Function *utah* MATLAB code, for calculating the chilling units according to the Utah model (Richardson et al., 1974)**

```matlab
function [ct] = utah(temperature)

ct = zeros(length(temperature),1);

%if 1.4<Temperature<=2.4
x = temperature>1.4 & temperature<=2.4;
ct(x) = ct(x)+0.5;

%if 2.4<Temperature<=9.1
x = temperature>2.4 & temperature<=9.1;
ct(x) = ct(x)+1;

%if 9.1<Temperature<=12.4
x = temperature>9.1 & temperature<=12.4;
ct(x) = ct(x)+0.5;

%if 12.4<Temperature<=15.9 multiplies by 0

%if 15.9<Temperature<=18
x = temperature>15.9 & temperature<=18;
```

```
ct(x) = ct(x)-0.5;

%if Temperature>18
x = temperature>18;
ct(x) = ct(x)-1;
```

**Code 3S. Function *north caroline* MATLAB code, for calculating the chilling units according to the North Caroline model (Shaltout and Unrath, 1983)**

```
function [ct] = noth_caroline(temperature)

% North Caroline model (from Shaltout and Unrath 1983, pg 959)

ct = zeros(length(temperature),1);
%if -1.1=<Temperature, zero contribution

%if -1.1<Temperature<=1.6
x = temperature>-1.1 & temperature<=1.6;
ct(x) = ct(x)+0.5;

%if 1.6<Temperature<=7.2
x = temperature>1.6 & temperature<=7.2;
ct(x) = ct(x)+1;

%if 7.2<Temperature<=13
x = temperature>7.2 & temperature<=13;
ct(x) = ct(x)+0.5;

%if 13<Temperature<=16.5, zero contribution

%if 16.5<Temperature<=19
x = temperature>16.5 & temperature<=19;
ct(x) = ct(x)-0.5;

%if 19<Temperature<=20.7
x = temperature>19 & temperature<=20.7;
ct(x) = ct(x)-1;

%if 20.7<Temperature<=22.1
x = temperature>20.7 & temperature<=22.1;
ct(x) = ct(x)-1.5;

%if 22.1>Temperature
x = temperature>22.1;
ct(x) = ct(x)-2;
```

**Code 4S. Function *melo_abreu* MATLAB code, for calculating the chilling units according to the de Melo-Abreu model (De Melo-Abreu et al., 2004)**

```matlab
function [ct] = melo_abreu(temperature)

% Model 1 for Olives (Melo-Abreu, fig. 1 and section 3.1.)

ct = zeros(length(temperature),1);

%Define constants
%To: optimum temperature for chilling (°C)
To=7.3;
%Tx: breakpoint temperature (°C)
Tx = 20.7;
%a: chilling units nullify when temperature is above Tx
a = -0.56;

%if 0<Temperature<=To
x = temperature>0 & temperature<=To;
ct(x) = ct(x)+temperature(x)/To;

%if To<Temperature<=Tx
x = temperature>To & temperature<=Tx;
ct(x) = ct(x) + 1 - (temperature(x)-To)*(1-a)/(Tx-To);

%if Temperature>Tx
x = temperature>Tx;
ct(x) = ct(x) + a;
```

**Code 5S. Function *hour_temp* MATLAB code for calculating hourly temperature from maximum and minimum daily temperatures following the de Wit et al. (1978) approach.**

```matlab
function [thour,time] = hour_temp(Tmax,Tmin,dates,lat)

% Calculates the temperature for each hour of the day taking into account
% the maximum and minimum temperatures

% Input parameters
    % Tmax: vector with maximum temperatures for each day
    % Tmin: vector with minimum temperatures for each day
    % dates: vector with the date for each row
    % lat: latitude of the cell

% Output parameters
    % thour: temperature for each hour
    % time: vector with the date and time for each row
```

```matlab
    %constants
    hour_col = 4;
    day_col = 3;
5   month_col = 2;
    year_col = 1;

    %creates the array of days, with values from 1 to 366
    d = zeros(length(dates),1);
10  for i =1 : length(dates)
        %it restarts the count with each new year
        if(dates(i,month_col)==1 && dates(i,day_col)==1)
            d(i) = 1;
        else
15          d(i) = d(i-1)+1;
        end
    end

    %creates Tmax_aux and Tmin_aux, as auxiliar variables for calculations
20  Tmax_aux = zeros(length(Tmax)+1,1);
    Tmax_aux(1) = Tmax(1);
    Tmax_aux(2:end) = Tmax;
    Tmin_aux = zeros(length(Tmin)+1,1);
    Tmin_aux(end) = Tmin(end);
25  Tmin_aux(1:(end-1)) = Tmin;

    % gets the sunrise hour (tr) for each day (d)
    [tr] = sunrise_time(d,lat);

30  % creates the array for saving hour temperatures
    thour = zeros(23,length(d));
    hour = zeros(23,length(d));
    years = zeros(23,length(d));
    months = zeros(23,length(d));
35  days = zeros(23,length(d));

    %gets the temperature of all days for each hour (0-23)
    for t=0:23

40      %if t<tr
        x = find (t<tr);
        if ~isempty(x)
            thour(t+1,x) = (Tmax_aux(x) + Tmin(x))/2 + ...
                (Tmax_aux(x)-Tmin(x))/2.*cos(pi*(t+10)./(tr(x)+10));
45      end
        %if t>=tr && t<14
        x = find(t>=tr & t<14);
        if ~isempty(x)
        thour(t+1,x) = (Tmax(x) + Tmin(x))/2 - ...
50          ((Tmax(x)-Tmin(x))/2).*cos(pi*(t-tr(x))./(14-tr(x)));
        end
```

```
      %if t>=14 && t<24
      if t>=14
            thour(t+1,:) = (Tmax + Tmin_aux(2:end))/2 + (Tmax-
  Tmin_aux(2:end))/2.*cos(pi*(t-14)./(tr+10));
      end
      hour(t+1,:) = t*ones(length(d),1);
      years(t+1,:) = dates(:,year_col);
      months(t+1,:) = dates(:,month_col);
      days(t+1,:) = dates(:,day_col);
end

%transforms the thour array into a vector
thour = thour(:);
time = zeros(length(thour),6);
time(:,year_col) = years(:);
time(:,month_col) = months(:);
time(:,day_col) = days(:);
time(:, hour_col) = hour(:);
```

**Code 6S. Function *sunrise_time* MATLAB code for calculate the sunrise time for each day from latitude**

```
function [h] = sunrise_time(d,Lat)

% calculates the sunrise time for each day (d) taking
% into account the latitude

% Input parameters
    % d: matrix with the days of year
% Output parameters
    % h: sunrise time for each day in the input matrix

%D: day of year in degrees
D = 360*d/365;
%dec: declination in degrees
dec = -23.5*cosd(D+9.865);
%w: angle at sunrise in degrees
w=acosd(-tand(Lat)*tand(dec));
%h: solar time in hour
h = 12-w/15;
```

**Code 7S. Function *calculate_period* MATLAB code for calculate initial and final chilling accumulation periods used for all chilling models but the Dynamic one.**

```
% This function takes the chilling accumulation and checks the "area"
% that would remain taking the maximum values (relatives). It is like if we
```

```matlab
    % trace a horizontal line, see where it intersects and we fill the
    % resulting area. The biggest area is the chosen one. Then, the minimum
    % is identified as the one laying in that region.

5   function [ pos_minimo, pos_maximo ] = calculate_period(d)

    acc = d;
    dlen = length(d);

10  % Remove the first part where only go down
    baja = 1;
    c = 0;
    while baja>0
        c = c + 1;
15
        % If arrives to the end means it never went down
        if c == dlen
            c = 1;
            baja = 0;
20          break;
        end

        % Does not go down
        if d(c+1)>d(c)
25          baja = 0;
        end
    end
    for i=1:(c-1)
        d(i) = NaN;
30  end

    x = isnan(d);
    numnan = length(d(x));
35  d = d(~x);

    % ListaEnlazada is just a linked list data structure
    CORTEs = ListaEnlazada();
40  MAXs = ListaEnlazada();
    AREAs = ListaEnlazada();

    anterior = d(1);
    for i=2:(dlen-numnan)-1
45
        if is_maximum(d,i)>0

            POs = find_left_cut(acc,d(i),i+numnan);

50          corte = 1;
            if length(POs)>0
```

```matlab
            % Remove the same point
            for j=1:length(POs)
                if POs(j)==(i+numnan)
                    POs(j) = NaN;
                end
            end
            x = isnan(POs);
            POs = POs(~x);

            if length(POs)==0
                corte = 1;
            else
                posaux = POs;
                for aa=1:length(POs)
                    pini = POs(aa);
                    pfin = i+numnan;

                    aux = acc(pini:pfin);
                    aux = max(aux);

                    % It does not count because it cut the graph
                    if acc(pini)<aux
                        x = posaux == POs(aa);
                        posaux = posaux(~x);
                    end
                end
                POs = posaux;

                dist = POs-(i+numnan);

                x = dist == min(dist);
                corte = POs(x);
                if length(corte)>1
                    corte = corte(1);
                end
            end
        end

        p1 = corte;
        p2 = i+numnan;

        area = 0;
        for j=p1:p2
            area = area + abs(max(d(i),acc(j))-min(d(i),acc(j)));
        end

        % Add to the end of the linked list
        MAXs.insertaFinal(i+numnan);
        CORTEs.insertaFinal(p1);
        CORTEs.insertaFinal(p2);
        AREAs.insertaFinal(area);
```

```matlab
            close all
        end

        anterior = d(i);
    end

    % Get number matrix from linked list
    MAXs = MAXs.getMatrizNumerica();
    CORTEs = CORTEs.getMatrizNumerica();
    AREAs = AREAs.getMatrizNumerica();

    if length(MAXs)>0

        x = AREAs == max(AREAs);
        p = find(x);
        if length(p)>0
            p=p(1);
        end

        % Search for the minimum
        posmin = CORTEs((p-1)*2+1);
        minim = acc(CORTEs((p-1)*2+1));
        for j=(CORTEs((p-1)*2+1)):(CORTEs((p-1)*2+2))
            if acc(j)<minim
                minim = acc(j);
                posmin = j;
            end
        end

        pos_minimo = posmin;
        pos_maximo = MAXs(p);

    else
        pos_minimo = NaN;
        pos_maximo = NaN;
    end

end
```

**Code 8S. Function *is_maximum* MATLAB code, auxiliary for *calculate_period* function**

```matlab
% Given an array with values, it tell us if the value in the position i is a
% relative maximum. It will be considered also maximum if it is the last one of
% a series of the same values

function r = is_maximum( d, i )
```

```matlab
    if i<=1
        r = 0;
        return;
    end

    ok = 0;
    if i==length(d)
        ok = 1;
    else
        if d(i)>d(i+1)
            ok = 1;
        end
    end

    % If its the last one or is bigger than the next one it can be maximum
    if ok>0
        if d(i)>d(i-1)
            r = 1;
            return;
        end
        aux = i;
        while d(aux) == d(i)
            aux = aux - 1;
        end
        if d(i)>d(aux)
            r = 1;
            return;
        else
            r = 0;
            return;
        end

    else
        r = 0;
        return;
    end

    end
```

**Code 9S. Function *find_left_cut* MATLAB code, auxiliary for *calculate_period* function**

```matlab
    % It finds the positions x where the plot intersects with the horizontal line y = v
    % (only cuts at the left of x) y = f(x)

    function [ POs ] = find_left_cut(d,v,x)

    POs = ListaEnlazada();
    for i=1:length(d)-1
```

```
     if   (d(i) > v && d(i+1) < v) || ...
               (d(i) < v && d(i+1) > v)
          POs.insertaFinal(i);
          POs.insertaFinal(i+1);
     else
          if d(i) == v
              POs.insertaFinal(i);
          else if d(i+1) == v
              POs.insertaFinal(i+1);
              end
          end
     end
end

% Delete repeated
POs = POs.eliminaRepetidos();

aux = x-1;
while aux>0 && d(aux)==d(x)
     posaux = POs.posiciones(aux);
     posaux = posaux.getMatrizNumerica();
     posaux = posaux(1);

     % Delete element in position posaux
     POs.eliminaN(posaux);
     aux = aux - 1;
end

% Convert to numerical matrix and we left only the left ones
POs = POs.getMatrizNumerica();
xxx = POs<x;
POs = POs(xxx);
```

---

## Editor Decision (ED1)

**Guest editor report: nhess-2018-392; original title "Chilling accumulation in temperate fruit trees in Spain under climate change"**

The authors have undertaken an ambitious research in assessing winter chill across Spain, as derived from meteorological observations and climatic projections. Each of the two reviewers have provided an excellent and detailed revision of the manuscript, to which the authors have responded in a detailed manner.

The two reviewers reach consensus on a number of critical points. The interactive comments from both reviewers have been well documented and the authors have formulated solutions to take the manuscript further in two separate documents (RC1 and RC2). I agree with the solutions presented by the authors. Below I highlight some points of attention for the authors.

Overall the research can be documented better in the manuscript such that justice is done to the rigorous work undertaken. The processing of meteorological observations and climate scenarios, and their relation to the impact on the Spanish fruit trees uses state-of-the-art methodology. Therefore I would suggest that the authors revise the manuscript according to their documentation and replies to the reviewers.

The following major points require the authors' attention:

1. Avoid vague descriptions and formulate more precisely what has been done, certainly in the abstract. [an example: "near and far future" is vague; define the periods "2021-2050" and "2071-2100"] Overall a focus on precise findings will improve the readability of both manuscript and abstract.
2. A comprehensive review of chilling requirements for different species will be of enormous relevance and interest to an international audience. To this extent, the authors' suggestion of adding a table is excellent. References to the literature, as already documented in the authors' replies to the reviewers, could be extended to include research that is relevant to Spain or similar climatic environments (e.g. California).
3. An important outcome of the research relates to winter chill reduction. It would be useful to discuss the number of times chilling requirements are compromised for the different periods studied.
4. The choice of keeping the different chill model results separately is underpinned by the reviewers' preference for the dynamic model, and therefore I recommend to keep the results separately as currently done. Nevertheless, a better documentation of the different chill models and temperature thresholds will clarify the comments made.
5. I leave it to the authors to decide whether to share their code in the supplementary material or document the formulas used.

Since most of the above points have been documented in the replies to the reviewers, the revised manuscript can be reviewed by the handling editor.